# Rational design of microRNA-responsive switch for programmable translational control in mammalian cells

Hui Ning [1,3], Gan Liu [2,3], Lei Li[1], Qiang Liu[2], Huiya Huang[2] & Zhen Xie [1] ✉

Artificial RNA translation modulation usually relies on multiple components, such as RNA binding proteins (RBPs) or microRNAs (miRNAs) for off-switches and double-inverter cascades for on-switches. Recently, translational circular RNAs (circRNAs) were developed as promising alternatives for linear messenger RNAs (mRNAs). However, circRNAs still lack straightforward and programmable translation control strategies. Here, we rationally design a programmable miRNA-responsive internal ribosome entry site (IRES) translation activation and repression (PROMITAR) platform capable of implementing miRNA-based translation upregulation and downregulation in a single RNA construct. Based on the PROMITAR platform, we construct logic gates and cell-type classifier circRNAs and successfully identify desired mammalian cell types. We also demonstrate the potential therapeutic application of our platform for targeted cancer cell killing by encoding a cytotoxic protein in our engineered circRNAs. We expect our platform to expand the toolbox for RNA synthetic biology and provide an approach for potential biomedical applications in the future.

RNA-based therapeutics, such as antisense oligonucleotides (ASO)[1], small interfering RNA (siRNA)[2], and messenger RNA (mRNA) vaccines[3], have shown great potential in biomedical applications. Compared with linear mRNA, circular RNA (circRNA) exhibits superior stability and extended duration in mammalian cells due to higher resistance to RNase degradation and lower immunogenicity[4,5]. Therefore, circRNA was considered a promising alternative for linear mRNA. To date, synthetic biologists have developed strategies to modulate gene expression at the RNA level in linear mRNA[6,7] and recently in circRNA[8]. These strategies usually included RNA binding protein (RBP) aptamers and microRNA (miRNA) binding sites (miRBSs) in response to RBP and miRNA to switch off translation and employed double-inverter cascades of off-switch modules to switch on translation. Therefore, multiple RNA constructs were needed to develop functional devices such as logic gates and cell-type classifiers, which increased the size of RNA circuits and brought in exogenous proteins with potential

immunogenicity. Several cleavage-based miRNA-responsive on-switches were reported previously[9–11], and these methods relied on miRNA cleavage to release poly adenosine tail[9], eliminate the degradation domain of linear mRNA[10], or eliminate the translation suppression sequence[11]. However, these cleavage-based miRNA-responsive on-switches are not applicable in circRNAs, because any cleavage on circRNAs would break down circRNAs and result in degradation of circRNAs. The strategy of cleavage-independent miRNA-responsive on-switches was reported previously[12], which relied on hindering the cap-initiated ribosome scanning by sequestering the Kozak sequence to turn off the translation. While the binding of target miRNA enables the release of the Kozak sequence and the cap-initiated scanning ribosome could recognize and bind to the Kozak sequence to turn on the translation. However, as the translation initiation of circRNA is cap-independent, this strategy might not be feasible in circRNAs. Unlike mRNA, circRNA uses the internal ribosome

[1]MOE Key Laboratory of Bioinformatics and Bioinformatics Division, Center for Synthetic and Systems Biology, Department of Automation, Beijing National Research Center for Information Science and Technology, Tsinghua University, Beijing 100084, China. [2]Syngentech Inc., Zhongguancun Life Science Park, Changping District, Beijing 102206, China. [3]These authors contributed equally: Hui Ning, Gan Liu. ✉e-mail: zhenxie@tsinghua.edu.cn

entry site (IRES) to initiate translation. IRES possesses a sophisticated RNA structure that interacts with translation initiation factors or ribosomes directly. It has been reported that engineering specific RNA structures, such as eToehold[13] and aptamers[14], in IRES sequences can regulate translation in response to RNA transcripts or small molecule ligands. However, the number of high-affinity aptamers for ligand-responsive IRES is limited, which hampers the programmability of this strategy. The type IV IRESs used in the eToehold exhibited modest expression levels in mammalian cells, and the ON/OFF ratio of mean fluorescence intensity of the eToehold was relatively low when it was used to sense endogenous mRNA transcripts.

Here, we report a strategy for the rational design of a programmable miRNA-responsive IRES translation activation and repression (PROMITAR) platform capable of implementing miRNA-based translation upregulation and downregulation in a single RNA construct. Based on the PROMITAR platform, we construct logic gates and cell-type classifier circRNAs and successfully identify desired mammalian cell types based on cell-type specific miRNA profiles. We also demonstrate the potential biomedical application and the versatility of our platform by performing targeted cancer cell killing and extending the design principle to another IRES, respectively.

## Results

### Construction of miRNA-responsive IRES translation activators (MITAs) based on hepatitis C virus (HCV) IRES

We first chose the hepatitis C virus (HCV) IRES to demonstrate our strategy because the structure of HCV IRES has been solved[15]. HCV IRES is composed of four domains. Previous literature has pointed out that a pseudoknot in domain IV played a crucial role in HCV IRES-mediated translation initiation, and the pseudoknot is sterically adjacent to the 5' end of HCV IRES[16]. Thus, we hypothesized that the native structure of HCV IRES would be distorted by inserting an upstream sequence (termed IRES structure distorting sequence, DS) complementary to the pseudoknot region, leading to inhibition of IRES-mediated translation. We generated a bicistronic fluorescence reporter construct to evaluate the translation efficiency of engineered IRES in human embryonic kidney (HEK293) cells using flow cytometry (Fig. 1a). The translation of EYFP is cap-dependent and serves as an internal control. The translation of tagBFP is IRES-initiated, and the ratio of mean fluorescence intensity (MFI) of tagBFP to EYFP denotes the IRES-mediated translation efficiency. In comparison with unmodified native IRES (nIRES), the MFI of tagBFP decreased when the length of DS exceeded 12 nucleotides and was reduced by 10-fold when the length of DS reached 20

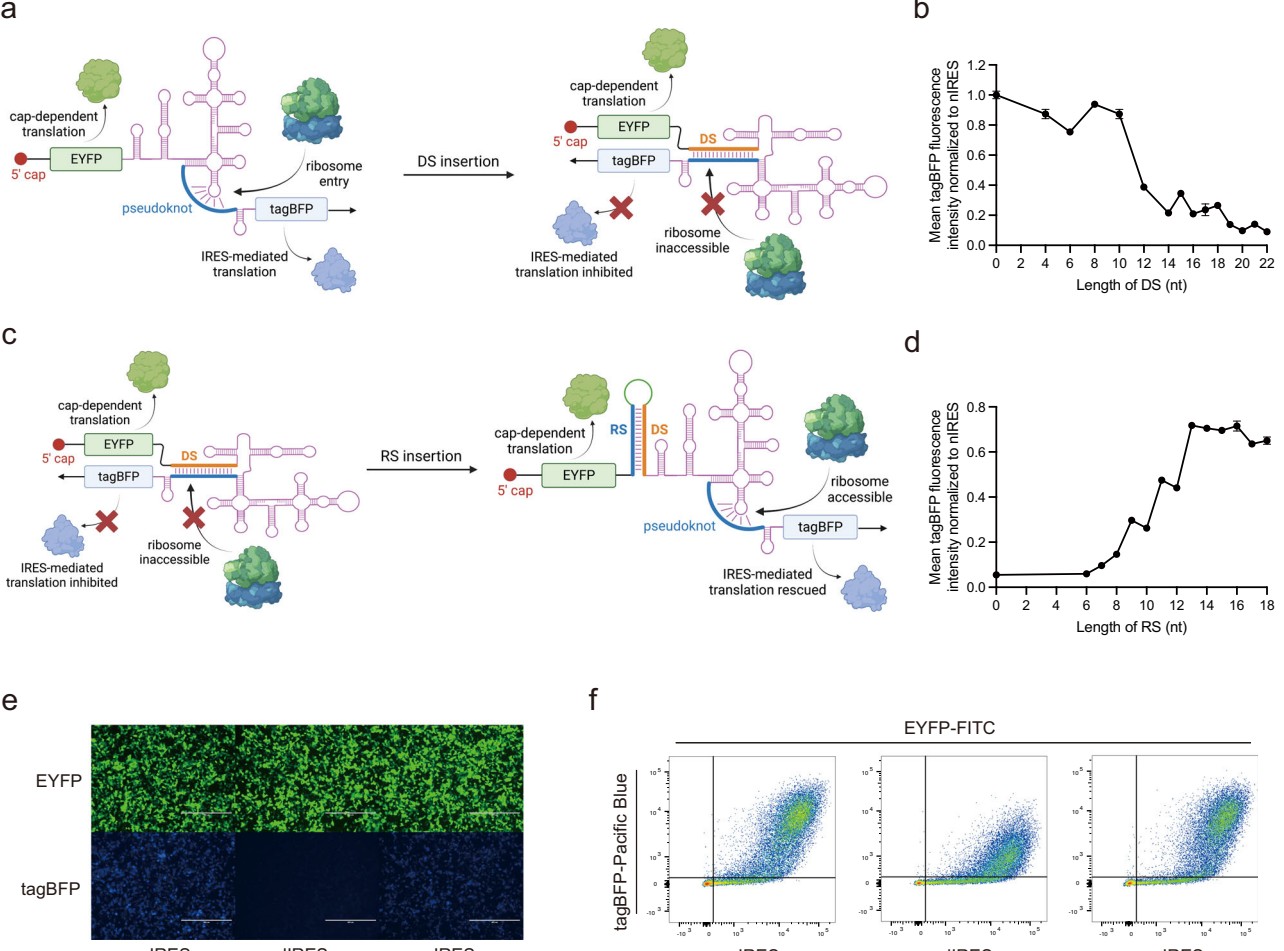

**Fig. 1 | Artificial control of HCV IRES-mediated translation through structure modulation. a** Schematic illustration of DS inhibition effect of the IRES translation activity. **b** Reduction of IRES-mediated tagBFP translation with different length of DS. **c** Schematic illustration of RS recovery effect of the IRES translation activity. **d** Recovery of IRES-mediated tagBFP translation with different length of RS. **e** Representative microscopy images of nIRES, dIRES, and rIRES. Each experiment was repeated three times independently with similar results. **f** Representative flow cytometry scatter plots of nIRES, dIRES, and rIRES. Each experiment was repeated

three times independently with similar results. HEK293 cell line was used for the transfection experiments. Data are presented as mean values with error bars representing the standard deviation of three independent biological replicates (*n* = 3 in each group). DS, distorting sequence. RS, rescuing sequence. nIRES, native IRES. dIRES, nIRES with 20-nt DS. rIRES, dIRES with 13-nt RS. Schematic illustration figures were created with BioRender.com with publication licenses. Source data are provided as a Source Data file.

nucleotides (termed this structure dIRES, Fig. 1b, Supplementary Fig. 1a–b).

Next, we designed and inserted a rescuing sequence (RS) complementary to the 20-nt DS, hypothesizing that base-pairing between RS and DS would restore the native structure and translation initiation function of HCV IRES (Fig. 1c). By introducing a RS upstream of DS with a 12-nt spacer, MFI of tagBFP increased when the length of RS exceeded 8 nucleotides and recovered to over 70% of nIRES when the length of RS reached 13 nucleotides (termed this structure rIRES, Fig. 1d–f, Supplementary Fig. 1c–e). Additionally, we also observed similar reduced tagBFP translation of dIRES and recovery of rIRES in human hepatocellular carcinoma (Huh7) cells as in HEK293 cells (Supplementary Fig. 1f, g).

To engineer programmable miRNA-responsive IRES translation activators (MITAs), we sought to design secondary structures with miRNA binding sites (miRBS) to sequester RS in a relatively stable structure. To ensure the miRNA-Argonaute (Ago) complex only binds to miRBS without cleavage, we designed miRBS as miRNA sponge[17] with 2-nucleotide mismatches with miRNA at positions 10 and 11. Enlighten by toehold switch[18] and other riboregulator design[19], we constructed three types of MITAs (Fig. 2a–c), namely toehold-like (TL, Fig. 2a), stem-loop (SL, Fig. 2b), and 3-arm-junction (AJ, Fig. 2c). We hypothesized that, in the absence of complementary miRNA, the 13-nt RS was sequestered in the designed structure, and IRES-mediated tagBFP translation was inhibited by the 20-nt DS (OFF state). While in the presence of complementary miRNA, the miRNA-Ago complex binds to the designed miRBS and interrupts the local secondary structure, releasing the 13-nt RS to recover IRES structure and translation activity (ON state). We first designed miRBS complementary to an artificial miR-FF4 and tested all three types of MITA in HEK293 with miRNA mimics co-transfection. Since more base-pairing in the RS-sequestering structure would stabilize the OFF-state configuration while more mismatches would disfavor the OFF-state configuration, we variated the number of base-pairing and mismatches to optimize the MITA structure. We designed a variety of engineered IRES structures, including toehold-like structures with varying base-pairing (TL14 to TL18, Fig. 2d, Supplementary Fig. 2a, b), stem-loop structures with different mismatches in stems (SL0, SL2 to SL6, Fig. 2e, Supplementary Fig. 2c, d), and 3-arm-junctions with different mismatches in junctions (AJ0, AJ2 to AJ5, Fig. 2f, Supplementary Fig. 2e, f). We transfected the miRNA mimics at low (10 nM), medium (30 nM), and high (50 nM) concentrations. All three types of MITA exhibited reduced IRES-mediated tagBFP translation activity with control mimics co-transfection, indicating the MITAs were in the OFF state in the absence of the desired miRNA. At medium and high transfection concentrations of miR-FF4 mimics, the tagBFP translation activity mostly recovered, indicating the presence of miR-FF4 triggered the MITAs to the ON state. Notably, the results also unveiled a dose-dependent activity of the designed MITAs, validating the miRNA sensor functionality of our MITA design. To confirm that miRNAs did not affect the basal IRES expression, we included control mimics, miR-FF4 mimics, and miR-21 mimics to test the influence of miRNA on IRES itself (Supplementary Fig. 2g). These control experiments demonstrated that miRNA mimics did not affect the basal expression of IRES (nIRES, dIRES, and rIRES). To further verify the specificity of our MITA design, we co-transfected our miR-FF4 responsive MITAs with either control mimics or non-target miR-199a mimics. These control experiments showed that the miR-FF4 responsive MITA was not responsive to non-target miRNA mimics, which confirmed the specificity of our MITA design (Supplementary Fig. 2h). Taken together, these results indicated that the translation activity of HCV IRES could be modulated by changing the conformation of IRES structure, and the conformational change could be designed to be miRNA-responsive.

## Construction of logic gates and cell-type classifiers based on the PROMITAR platform

Having demonstrated the strategy of programming MITAs in response to specific artificial miRNA mimics, we then sought to engineer MITAs to sense dual miRNA mimics to implement logic gates. We tandemly ligated two TL16 motifs for miR-FF4 and miR-199a (a miRNA that showed little expression in HEK293, Supplementary Fig. 3a) respectively, and tested in HEK293 with miRNA mimics co-transfection (Fig. 3a, b, Supplementary Fig. 3c, d). With either miR-FF4 or miR-199a mimics co-transfected, IRES-mediated tagBFP translation recovered, indicating that the tandem TL16 design could implement the OR logic function. We then designed the SL3 motif with both miR-FF4 and miR-199a binding sites (Fig. 3c, d, Supplementary Fig. 3e, f). With either miR-FF4 or miR-199a mimics co-transfected, tagBFP translation was inhibited as the negative control. With both miR-FF4 and miR-199a mimics co-transfected, tagBFP translation mostly recovered, indicating that the two-input SL3 design could serve as an AND logic gate. Furthermore, we also investigated the potential influence of miRBS arrangement on our system by altering the order of miRBS within OR-gates and AND-gates (Supplementary Fig. 3g, h). The results were consistent with our prior findings, indicating that our logic-gate design remained effective irrespective of the miRBS arrangement. To implement the NOT logic function, we generated a programmable miRNA-responsive IRES translation repressor (MITR) by directly inserting fully complementary binding sites of miR-FF4 both upstream and downstream of the nIRES (Fig. 3e, f, Supplementary Fig. 3i, j). The MITR exhibited a slightly lower translation level compared with nIRES in the absence of miR-FF4. With miR-FF4 co-transfected, the tagBFP translation was significantly inhibited, indicating that MITR could implement the NOT logic function.

After characterizing our MITA and MITR design with exogenous miRNA mimics, we then sought to employ MITA to sense endogenous miRNA. We designed miRBS complementary to miR-21, a miRNA that is highly expressed in most cancer cells, such as Huh7 but not in HEK293 (Supplementary Fig. 3a, b). Since IRES usually differs in the basal expression among different cell lines[4], and we indeed found that HCV IRES showed 1.2–1.5 times higher expression in Huh7 than in HEK293 when these cells were transfected with our bicistronic reporter plasmids (Supplementary Fig. 1f, g). Hence, we normalized the reporter tagBFP expression by nIRES or rIRES expression in different cell lines to eliminate the inherent difference of IRES expression between these cell lines, which allowed us to exclusively illustrate the individual impact of our engineering strategies on IRES translation.

We then examined the normalized tagBFP translation of all three types of MITA in Huh7 and HEK293 cells (Fig. 3g, Supplementary Fig. 4a, b). As a result, all MITAs showed higher MFI of tagBFP in Huh7 than HEK293, and TL16, SL3 and AJ2 exhibited the highest fold changes (up to 3-fold). Interestingly, SL5, SL6, AJ4, and AJ5 showed higher MFI of tagBFP compared with rIRES, probably because the Ago proteins enhanced the translation initiation activity of HCV IRES as previously reported[20].

To rule out the possibility that the designed MITAs' structural changes could be induced by intracellular context instead of target miRNAs, we transfected each version of exogenous miR-FF4 responsive MITA plasmid constructs into HEK293 and Huh7 cells without the co-transfection of miR-FF4 mimics (Supplementary Fig. 4c). The differential expressions of MITAs comprising non-target miRBS (i.e., miR-FF4 binding sites) between the HEK293 and Huh7 were similar to the inherent IRES expression difference between HEK293 and Huh7, contrasting with the differential expression pattern of miR-21 responsive MITAs. These results indicated that our MITA design was not modulated by the cellular context. To verify the miRNA specificity of our miR-21 responsive MITAs design, we tested miR-21 responsive MITAs by introducing miR-21 inhibitors in Huh7 and miR-21 mimics in HEK293 (Supplementary Fig. 4d, e). The results showed a decrease in tagBFP

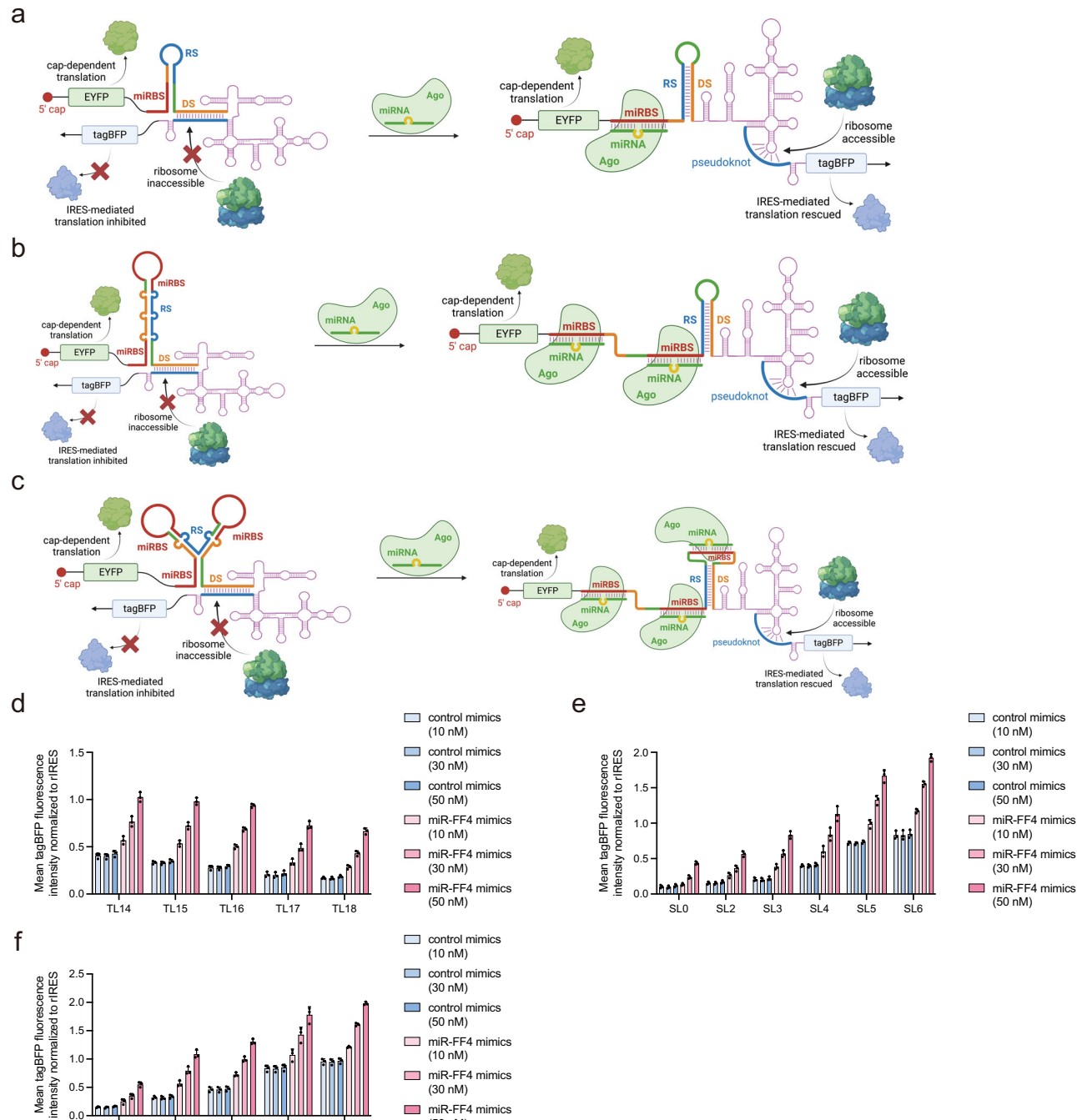

**Fig. 2 | Design and optimization of three types of MITA based on different secondary structures. a** Schematic illustration of Toehold-like design. **b** Schematic illustration of Stem-loop design. **c** Schematic illustration of 3-arm-junction design. **d** IRES-mediated tagBFP translation of TL design in response to artificial miR-FF4 mimics (TLx, x denotes the number of base-pairing in the stem). **e** IRES-mediated tagBFP translation of SL design in response to artificial miR-FF4 mimics (SLx, x denotes the number of mismatches in the stem). **f** IRES-mediated tagBFP translation of AJ design in response to artificial miR-FF4 mimics (AJx, x denotes the

number of mismatches in the 3-arm-junction). HEK293 cell line was used for the transfection experiments. Data are presented as mean values with error bars representing the standard deviation of three independent biological replicates ($n = 3$ in each group). miRBS, miRNA binding sites. Ago, argonaute. MITA, miRNA-responsive IRES translation activator. TL, toehold-like design of MITA. SL, Stem-loop design of MITA. AJ, 3-arm-junction design MITA. Schematic illustration figures were created with BioRender.com with publication licenses. Source data are provided as a Source Data file.

reporter expression with miR-21 inhibitor co-transfection in Huh7, and an increase in tagBFP reporter expression with miR-21 mimics co-transfection in HEK293. These results confirmed the specificity of our miR-21 responsive MITA design. In addition to Huh7 cells, we also tested the performance of miR-21-responsive MITAs in HeLa cells, another cancer cell line exhibiting high miR-21 activity[21] (Supplementary Fig. 4f). Consistent with our observations in Huh7 cells, miR-21

responsive MITAs also exhibited higher tagBFP reporter expression in HeLa cells than in HEK293 cells. To further explore our MITA design's capacity to respond to a diverse range of miRNA signals, we employed another endogenous miRNA, miR-18a, which showed high activity in HEK293 but low activity in Huh7 cells (Supplementary Fig. 3a, b). The results showed that the tagBFP reporter expression was higher in HEK293 cells than that in Huh7 cells (Fig. 3h, Supplementary Fig. 4g, h).

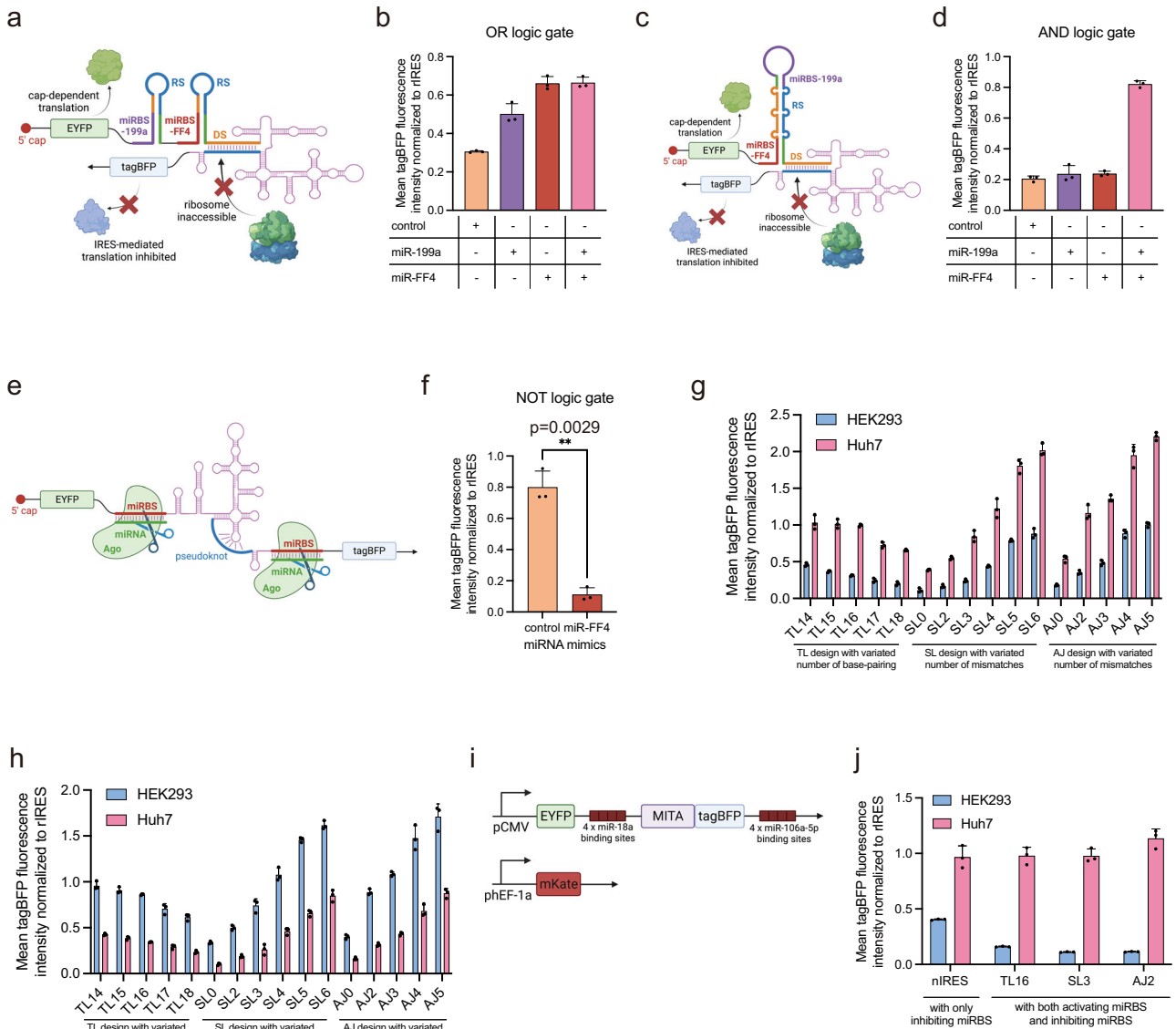

**Fig. 3 | Engineering miRNA-responsive IRES translation switches to implement logic gates and cell-type classifiers in mammalian cells. a–f** Implementing logic gates by programming MITA/MITR design. The IRES-mediated tagBFP fluorescence intensity was measured in HEK293 cells with miRNA mimics or control mimics co-transfection. **a** Schematic illustration of OR logic gate implemented by ligation of two TL16 modules for miR-199a and miR-FF4. **b** Tandem ligation of two different TL16 modules exhibited OR-logic computation function. **c** Schematic illustration of AND logic gate implemented by programming SL3 design for miR-199a and miR-FF4. **d** Integration of two different miRBS into SL3 design exhibited AND-logic computation function. **e** Schematic illustration of NOT logic gate (i.e., MITR) implemented by inserting miRBS fully complementary to miR-FF4. **f** Insertion of miRBS fully complementary to the input miRNA exhibited NOT-logic computation function. **g** Construction and optimization of miR-21 responsive MITAs design by variating the number of base-pairing or mismatches in designed structures. TL, SL, and AJ were all designed to sense endogenous miR-21. IRES-mediated tagBFP fluorescence intensity was measured in Huh7 (miR-21 high) and HEK293 (miR-21

low) cells. **h** Construction and optimization of miR-18a responsive MITAs design by variating the number of base-pairing or mismatches in designed structures. TL, SL, and AJ were all designed to sense endogenous miR-18a. IRES-mediated tagBFP fluorescence intensity was measured in Huh7 (miR-18a low) and HEK293 (miR-18a high) cells. **i** Schematic illustration of the construction of cell-type classifiers based on the PROMITAR platform. The mKate fluorescence protein was co-transfected as an internal control. **j** The MITA and MITR designs showed cooperating effects regarding cell-type classification. HEK293 and Huh7 cell lines were used for the transfection experiments. Data are presented as mean values with error bars representing the standard deviation of three independent biological replicates ($n = 3$ in each group). Statistical analysis of the results was performed by a two-tailed unpaired Welch's $t$-test, assuming unequal variances. *$p < 0.05$. **$p < 0.005$ ($p = 0.0029$ in Fig. 3f). miR, microRNA. pCMV, CMV promoter. phEF-1a, EF-1a promoter. Schematic illustration figures were created with BioRender.com with publication licenses. Source data are provided as a Source Data file.

Given the opposite miRNA activity levels of miR-18a and miR-21 in HEK293 and Huh7 cells, the reversed tagBFP differential expression patterns of corresponding designs of miR-18a-responsive and miR-21-responsive MITAs provided further evidence for the applicability of our MITA design for detecting diverse miRNA signals.

Combining the MITA and MITR designs, we then sought to employ our PROMITAR platform to construct cell-type classifiers. We

tested several miRNAs reported previously that showed high repression activity in HEK293 (Supplementary Fig. 3a, b)[22]. We chose miR-21 as Huh7-high activity miRNA to construct MITA modules, and miR-106a-5p and miR-18a as HEK293-high activity miRNAs to implement NOT logic gates. We constructed vectors containing miR-21 responsive TL16, SL3, or AJ2 MITA with 4 tandem repeats of HEK293-high miRBS upstream of the MITA and downstream of the tagBFP reporter (Fig. 3i).

We transfected each of these plasmid constructs into Huh7 and HEK293 cells with mKate fluorescence protein as an internal control. As a result, we observed up to a 7-fold increase of tagBFP fluorescence intensity in Huh7 compared with HEK293 (Fig. 3j). Notably, we only observed a ~2-fold increase of fluorescence signal when we employed MITA (Fig. 3g) or MITR (Fig. 3j). Collectively, these results indicated a cooperating effect between MITA and MITR in regard to cell-type classification. Since the basal expression of HCV IRES in Huh7 cells was ~1.2–1.5 times higher than that in HEK293 cells, combined with the higher normalized expression of MITA and classifiers in Huh7 relative to HEK293, we observed that MITA and classifiers showed much higher absolute expression in Huh7 cells (up to 9-fold, Supplementary Fig. 4i, j).

### Construction and validation of MITAs and cell-type classifiers in circular RNAs (circRNAs) for targeted cancer cell identification and elimination

To apply MITA modules and cell-type classifiers in circRNAs, we constructed plasmid templates containing permuted intron-exon (PIE)[4] for in vitro transcription (IVT) of circRNA (Fig. 4a). With firefly luciferase as a reporter, we first transfected circRNAs encoding nIRES, dIRES, and rIRES into HEK293 and Huh7 cells, respectively. We observed the same inhibition and recovery effect as plasmid transfections (Fig. 4b, Supplementary Fig. 5a). Next, we transfected circRNAs encoding TL16, SL3, or AJ2 miR-21 responsive MITA into Huh7 and HEK293 cells, respectively (Fig. 4c, Supplementary Fig. 5b). The TL16 or SL3 design exhibited up to 3-fold luminescence signal increase, and the AJ2 exhibited a 6-fold signal increase in Huh7 compared with HEK293. These results indicated that our MITA modules were also miRNA-activated in circRNA. Finally, we examined the cell-type classifiers in circRNA (Fig. 4d, e, Supplementary Fig. 5c). We observed up to 8-fold changes of luminescence signal in Huh7 compared with HEK293, indicating that our cell-type classifier circRNAs could accurately identify desired cancer cells, and might serve as potential cancer-targeting therapeutics. Notably, we observed that the basal expression of HCV IRES in Huh7 cells was ~3 times higher than that in HEK293 cells with circRNA transfection (Supplementary Fig. 5a), which meant that our design enabled much higher absolute expression in Huh7 (up to 9-fold for MITA, Supplementary Fig. 5b; up to 18-fold for classifier, Supplementary Fig. 5c), facilitating potential applications such as cell classification and selective killing of cancer cells. To further elucidate the potential therapeutic applications of our cell-type classifier circRNAs based on the PROMITAR platform, we replaced the luciferase reporter with the N-terminal section of Gasdermin D (GSDMD), a pore-forming protein that is involved in the inflammatory cell death pathway known as pyroptosis[23] (Fig. 4f). We then performed the lactate dehydrogenase (LDH) cytotoxicity assay to evaluate the cancer-cell-specific cytotoxicity of our MITA-containing GSDMD-encoding circRNAs. We observed a substantial difference in cytotoxicity levels, with an 18-fold higher LDH activity in Huh7 cells compared to HEK293 cells (Fig. 4g). Furthermore, we transfected the GSDMD-encoding cell-type classifier circRNA into Huh7 and HEK293 cells and observed an even higher difference in cytotoxicity (up to 89-fold, Fig. 4h, i, Supplementary Fig. 5d). To further evaluate the performance of our targeted cancer killing circRNAs in heterogeneous cell populations, we engineered two fluorescently tagged stable cell lines: Huh7 cells expressing EYFP and HEK293 cells expressing mScarlet (Supplementary Fig. 6). These cell lines were co-cultured, and subsequently transfected with lipid nanoparticles (LNPs) encapsulating either our cytotoxic GSDMD-encoding circRNA classifiers or noncytotoxic circRNAs as controls. Notably, we observed a substantial decrease in the ratio of surviving EYFP[+] Huh7 cell percentage to the surviving mScarlet[+] HEK293 cell percentage after transfection with the classifier circRNA (Fig. 4j, k). These

results indicated that the cell-type classifier functioned correctly and corroborated the functionality of our cell-type classifier in selectively inducing pyroptosis in Huh7 cells within mixed cellular populations. Collectively, these findings highlighted the potential applicability of our cell-type classifiers for targeted cancer therapy applications.

To evaluate the versatility of our cell-type classifiers, we sought to alter the miRNA binding sites (miRBSs) of the MITA and MITR modules in the cytotoxic GSDMD-encoding circRNA, while maintaining the optimized AJ2 secondary structure of the MITA module (generating classifiers 1 to 4, Supplementary Fig. 7a). We first swapped the miRBSs of the previous classifier (containing miR-21 MITA and miR-18a MITR) to generate a new classifier with miR-18a MITA and miR-21 MITR, expecting this to confer HEK293 cell-targeting cytotoxicity. We then performed LDH assays and co-culture flow cytometry experiments (Supplementary Fig. 7a, c, and e, f, Supplementary Fig. 8a, b, and d–g), and we observed reversed cell killing effects for this new classifier 2 compared with the original classifier 1, consistent with the altered miRBS order.

We further extended our classifier design principle for distinguishing Huh7 cells from another liver-derived line, L-02. We first identified let-7c[24] as a highly active L-02-specific miRNA marker using flow cytometry with both linear mRNA and circRNA transfection (Supplementary Fig. 7b). Both linear mRNA and circRNA transfection experiments showed that let-7c exhibited much higher activity in L-02 cells than Huh7 cells, with a more substantial repression activity of let-7c in circRNA transfection compared with linear mRNA transfection. With miR-21 as the Huh7-specific marker and let-7c as the L-02-specific marker, we designed two additional classifiers: classifier 3 with miR-21 MITA and let-7c MITR to distinguish Huh7 from L-02, and classifier 4 with let-7c MITA and miR-21 MITR to distinguish L-02 from Huh7 (Supplementary Fig. 7a). Transfection experiments showed classifier 3 induced stronger Huh7 cell death compared to L-02, while classifier 4 was more cytotoxic to L-02 versus Huh7 (Supplementary Fig. 7d and g, h, Supplementary Fig. 8b, c, and h–k). Collectively, these results demonstrated that our PROMITAR-based cell-type classifier design principle can distinguish cell types based on their unique miRNA expression profiles.

### Extending the design principles of the PROMITAR platform to classical swine fever virus (CSFV) IRES

To further demonstrate the versatility of our strategy, we then sought to expand the PROMITAR platform by applying our design principle to the classical swine fever virus (CSFV) IRES, which owns a similar pseudoknot structure as HCV IRES[25] (Fig. 5a). Firstly, we also designed a 20-nt DS targeting the pseudoknot of the CSFV IRES (also, termed this structure dIRES) and observed a similar inhibition effect as in HCV IRES (Fig. 5b, Supplementary Fig. 9a), suggesting that our strategy could extend to other IRES. Then we also introduced a series of RS upstream of the 20-nt DS and observed up to 70% recovery rate when the length of RS reached 13 nucleotides (Fig. 5c, d, Supplementary Fig. 9b, c). Following a similar approach as we designed the MITA and MITR with HCV IRES, we developed customized MITA and MITR modules based on the CSFV IRES by incorporating miRBSs responsive to miR-21 and miR-FF4, respectively. With the 20-nt DS and 13-nt RS, we designed RNA secondary structures with miR-21 binding sites sequestering the RS for MITAs as in HCV IRES, and designed MITR by inserting fully complementary binding sites of miR-FF4 both upstream and downstream of the CSFV IRES. We transfected the MITA-containing reporter plasmid into Huh7 and HEK293 cells (Fig. 5e, Supplementary Fig. 9d, e), and co-transfected MITR-containing reporter plasmids with miRNA mimics in HEK293 cells (Fig. 5f, g, Supplementary Fig. 9f, g). Notably, these constructs based on CSFV IRES performed in a manner closely analogous to the previously tested HCV IRES-based designs. Collectively, these results suggested that our design principle offered a

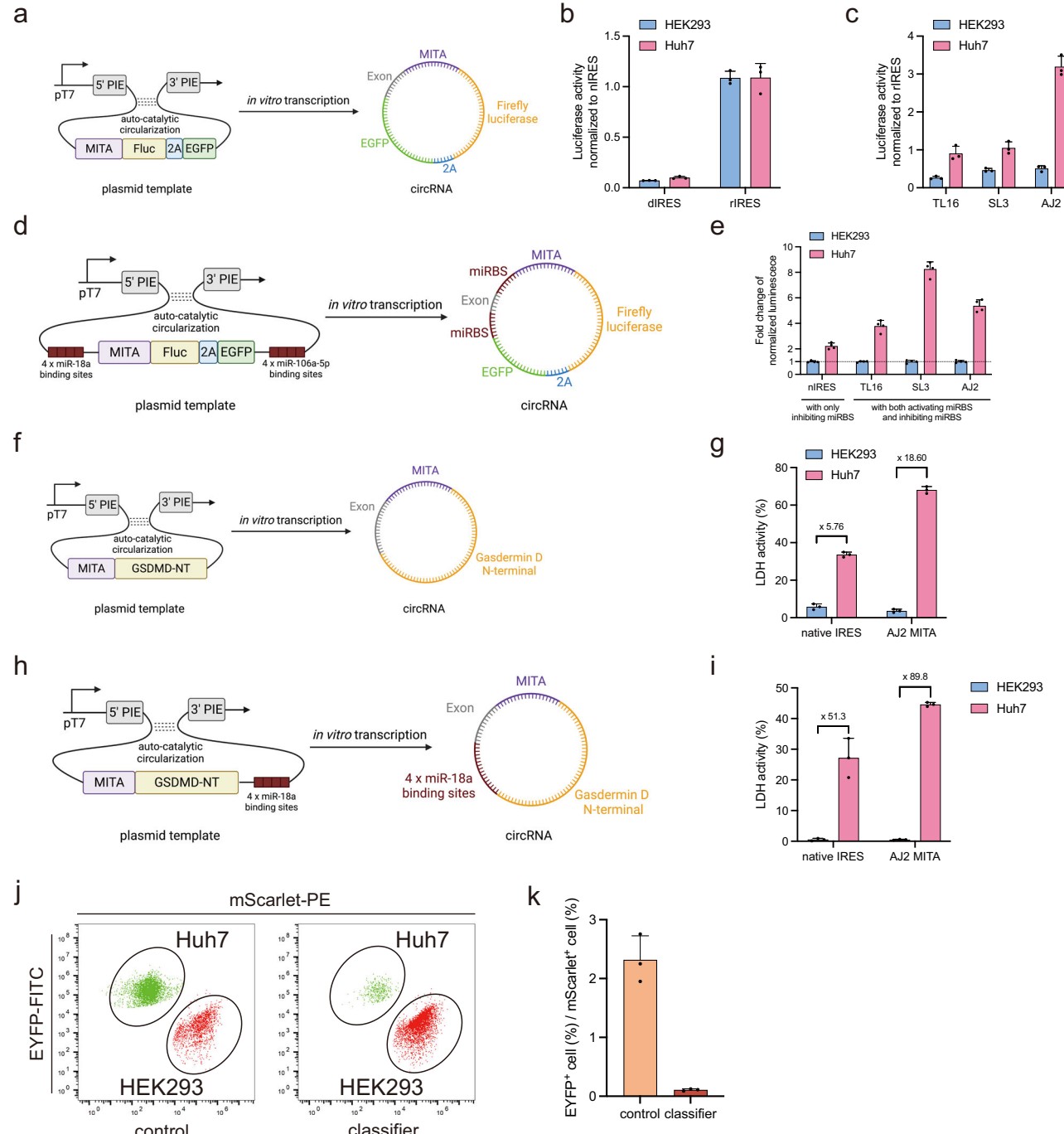

**Fig. 4 | Validation and application of the MITA modules and cell-type classifiers in circular RNAs. a** Schematic illustration of the plasmid constructs for in vitro transcription of circRNAs. **b** Characterization of nIRES, dIRES, and rIRES in circRNA in Huh7 and HEK293 cells. **c** Characterization of MITA modules in circRNA in Huh7 and HEK293 cells. MITA modules were designed for endogenous miR-21.
**d** Schematic illustration of the construction of cell-type classifiers in circRNA.
**e** Characterization of cell-type classifiers in circRNA. Fold change was calculated as the normalized luminescence of Huh7 divided by the normalized luminescence of HEK293. **f** Schematic illustration of the construction of MITA-containing cytotoxic GSDMD-encoding circRNA. **g** Characterization of the nIRES-containing and AJ2 MITA-containing cytotoxic GSDMD-encoding circRNA in Huh7 and HEK293 cells. Cytotoxicity was characterized by LDH activity measured by LDH cytotoxicity assay. **h** Schematic illustration of the construction of cytotoxic GSDMD-encoding cell-type classifier circRNA. **i** Characterization of the cytotoxic GSDMD-encoding

cell-type classifier circRNA in Huh7 and HEK293 cells. **j, k** Characterization of the AJ2 MITA-containing cytotoxic GSDMD-encoding circRNA in cell mixtures. Huh7 (EYFP[+]) cells and HEK293 (mScarlet[+]) cells were co-cultured and cytotoxic GSDMD-encoding (classifier) or noncytotoxic (control) circRNAs were transfected into the cell mixtures, respectively. **j** The scatter plots show the fractions of Huh7 (EYFP[+]) cells and HEK293 (mScarlet[+]) cells under the indicated conditions. **k** The bar chart shows the ratio of the surviving Huh7 (EYFP[+]) cell percentage to the surviving HEK293 (mScarlet[+]) cell percentage under the indicated conditions. HEK293 and Huh7 cell lines were used for the transfection experiments. Data are presented as mean values with error bars representing the standard deviation of three independent biological replicates ($n = 3$ in each group). pT7, T7 promoter. PIE, permuted-intron-exon. Fluc, firefly luciferase. 2 A, 2A-peptide. circRNA, circular RNA. Schematic illustration figures were created with BioRender.com with publication licenses. Source data are provided as a Source Data file.

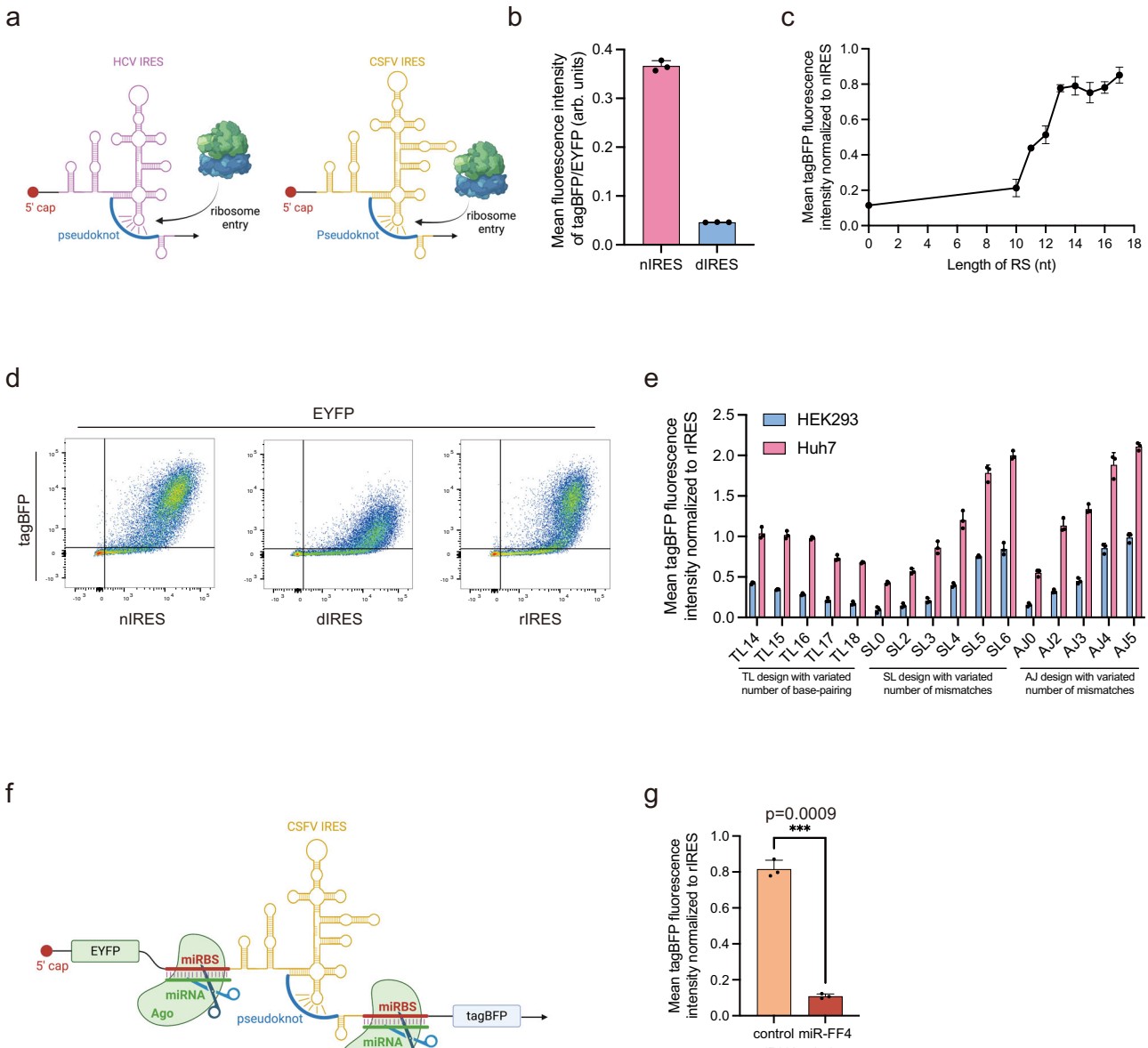

**Fig. 5 | Construction of the MITA and MITR modules with the CSFV IRES.**
**a** Schematic illustration of the similar secondary structures of HCV IRES and CSFV IRES. Both HCV IRES and CSFV IRES contained targetable pseudoknot structures adjacent to the translation initiation start codon. **b** Reduction of CSFV IRES-mediated tagBFP translation with a 20-nt DS. **c** Recovery of CSFV IRES-mediated tagBFP translation with different length of RS. **d** Representative flow cytometry scatter plots of nIRES, dIRES, and rIRES of CSFV IRES. Each experiment was repeated three times independently with similar results. HEK293 cell line was used for the transfection experiments in panel **b**–**d**. **e** Construction and optimization of miR-21 responsive CSFV MITAs by variating the number of base-pairing or mismatches in designed structures. TL, SL, and AJ were all designed to sense endogenous miR-21. CSFV IRES-mediated tagBFP fluorescence intensity was measured in Huh7 (miR-

21 high) and HEK293 (miR-21 low) cells. **f** Schematic illustration of the CSFV MITR designed by inserting miRBS fully complementary to miR-FF4. **g** Insertion of miRBS fully complementary to the input miRNA exhibited miRNA-responsive CSFV IRES translation repression function. HEK293 cell line was used for the transfection experiments. Data are presented as mean values with error bars representing the standard deviation of three independent biological replicates ($n = 3$ in each group). Statistical analysis of the results was performed by a two-tailed unpaired Welch's $t$-test, assuming unequal variances. *$p < 0.05$. **$p < 0.005$. ***$p < 0.001$ ($p = 0.0009$ in Fig. 5g). arb. units, arbitrary units. Schematic illustration figures were created with BioRender.com with publication licenses. Source data are provided as a Source Data file.

versatile framework for developing other PROMITAR platforms capable of engineering diverse IRES sequences while preserving the desired miRNA-responsive functionality.

## Discussion

In this study, we developed the PROMITAR platform, which enabled miRNA-dependent translation activation and translation repression in a single RNA construct. Based on the PROMITAR platform, we constructed logic gates and cell-type classifier circRNAs and successfully

identified desired mammalian cell types. The most important feature of the PROMITAR platform is the ability to simultaneously integrate miRNA-dependent translation activation and repression within one single RNA molecule, which - to the best of our knowledge - has not been reported in the literature previously. Furthermore, the successful identification of specific mammalian cell types and the precise targeting and killing of cancer cells using our cell-type classifier circRNAs indicated the potential therapeutic application of the PROMITAR platform. Therefore, our platform not only expanded the synthetic

biology toolbox for RNA translational control but also provided a potential approach for precision medicine in the future. It should be noted that the use of the IRES itself can lead to cell-type-dependent differences in basal translation level. Therefore, it might be another potential approach to construct cell-type classifiers by identifying IRESs with desired inherent expression biases. However, IRESs identified in this manner would be confined to specific cell lines in which these IRESs exhibited high activities, which would be unsuitable for customized cell-type classification or logic computation function. Moreover, amplifying these intrinsic cell-type differences of IRES via our PROMITAR system is also meaningful. For example, the basal LDH activity induced by HCV-IRES-mediated GSDMD expression showed a 5.8-fold difference between Huh7 and HEK293 cells, but introducing circRNA-ON regulation expanded this difference to 18.6-fold (Fig. 4g). Furthermore, although circRNA-OFF regulation expanded the cell-type difference to 51-fold, combining circRNA-OFF and circRNA-ON regulation amplified the difference to ~90-fold (Fig. 4i).

The programmability and versatility of the PROMITAR platform are also key facets. On the one hand, the PROMITAR platform established in this study can be modified to respond to different miRNAs, enabling the development of a wide variety of logic gates and cell-type classifiers. On the other hand, PROMITAR platforms based on other IRES can be developed and optimized following the design principle presented in this study. Although we used HCV IRES and another IRES with known pseudoknot structure to demonstrate our strategy in this study, for those IRES sequences with little knowledge of their structure-activity relationship, it is feasible to screen for the best DS by constructing a DS library targeting different regions of the IRES. Once the best target region and DS were identified, our design principle could be applied to engineer other PROMITAR platforms.

Combining with previously reported methods[6–8] of RNA-based circuit construction, we expect to employ our PROMITAR platform to construct sophisticated RNA circuits for programmable cellular functions, which promises potential therapeutic applications in the future. We also envisage the integration of the PROMITAR platform with other regulation modalities, such as protein-level regulation[26,27]. Integration with these additional regulatory layers could further enhance the specificity of gene expression control, yielding even more robust and sophisticated biological circuits. This would not only enhance the precision and control of cellular functions but could also facilitate the development of more advanced biomedical applications in the future.

## Methods

### Reagents and enzymes
Restriction endonucleases, ATP, polynucleotide kinase (PNK), T4 DNA ligase, and Q5 High-Fidelity DNA polymerase were purchased from New England Biolabs. Oligonucleotides were synthesized by Tsingke Co., Ltd. miRNA mimics and inhibitors were purchased from Beijing Syngentech Co., Ltd.

### Plasmid construction
When required, equal molar amounts of oligonucleotides were annealed in $1 \times$ PNK buffer by heating to 95 °C and gradually cooling down (−1 °C per min) to 37 °C, and then 1 μM of the annealed product was phosphorylated by 0.5 U/μL PNK in the presence of 0.5 mM ATP. Golden Gate Assembly was performed according to the protocols of New England Biolabs. The DNA sequences of plasmid constructs were listed in Supplementary Data 1 and the Benchling website: https://benchling.com/hui_ning/f_/K81GiYaI-rational-design-of-microrna-responsive-switch-for-programmable-translational-control-in-mammalian-cells/.

### Cell culture and transfection
HEK293 (HEK293FT) cell lines were purchased from Life Technologies. Huh7 cell line was purchased from BeNa culture collection Co., Ltd.

Hela and L-02 cell line was purchased from ATCC. HEK293, Huh7, and Hela cells were maintained in high-glucose DMEM (Invitrogen) supplemented with 10% FBS (DiNing) and 1% penicillin-streptomycin mix (Invitrogen) at 37 °C, 100% humidity, and 5% $CO_2$. L-02 cells were maintained in RPMI-1640 (Invitrogen) supplemented with 10% FBS (DiNing) and 1% penicillin-streptomycin mix (Invitrogen) at 37 °C, 100% humidity, and 5% $CO_2$. Lipo8000 ((Beyotime, Shanghai, China)) was used for plasmids and miRNA mimics transfection. Transfection was performed according to the manufacturer's protocol. Briefly, one day before transfection, ~3 × 10^5 cells in 1 mL DMEM complete media were seeded into each well of the 24-well plate (Corning). For plasmid transfection, 0.6 μg of plasmids were mixed with 0.6ug Lipo8000 and then transfected into cells in each well. For miRNA mimics or miRNA inhibitor co-transfection, miRNA mimics or miRNA inhibitors were mixed with 0.6 μg of plasmids and 0.6ug Lipo8000 at a final transfection concentration of 50 nM (or indicated concentration), and then transfected into cells in each well. For circRNA transfection, circRNAs were encapsulated by lipid nanoparticles as described below and transfected into cells with 0.6 μg per well. Lentiviruses (lenti-LW444 from Beijing Syngentech Co., Ltd.) were used for the construction of stable cell lines. Transfection was performed according to the manufacturer's protocol. The EYFP-Huh7 cell line was derived from Huh7 cell line by transfection with the lenti-GF1-CMV-EYFP lentivirus. The mScarlet-HEK293 cell line was derived from HEK293 cell line by transfection with the lenti-GF1-CMV-mScarlet lentivirus.

### Circular RNA production and purification
The production and purification of circular RNA (circRNA) was performed according to a previous report[4]. Briefly, the DNA templates for circRNA precursors were generated by linearizing plasmid constructs using Xba I endonuclease. The circRNA precursors were produced by in vitro transcription (IVT) from the DNA templates with T7 Polymerase (Novoprotein). After IVT, the RNA products were treated with DNase I (Thermo Fisher Scientific) for 30 min to digest the DNA templates. After DNase I digestion, GTP (Novoprotein) was added to the RNA products at a final concentration of 2 mM, and the RNA products were incubated at 55 °C for 15 min to catalyze the cyclization of circRNAs. Then, the circRNA products were treated with RNase R (Novoprotein) at 37 °C for 15 min to remove the precursor RNA. The RNase R-treated circRNA was purified by LiCl (Novoprotein).

### Lipid nanoparticle encapsulation of circRNA
The circRNAs were encapsulated with lipid nanoparticles (LNPs) prepared using a modified procedure of a previous report[28]. Briefly, Lipids were dissolved in ethanol containing an ionizable lipid (SM-102), 1, 2-distearoyl-sn-glycero-3-phosphocholine (DSPC), cholesterol, and PEG-lipid (with molar ratios of 50:10:38.5:1.5). The lipid mixture was combined with 50 mM citrate buffer (pH4.0) containing circRNA at a ratio of 1:3 through a Lipid Mixer (Micronano INano L). Formulations were then diafiltrated against 30x volume of PBS (pH7.4) through an Ultra Centrifugal Filter Unit (Millipore) with 100 kD molecular weight cutoffs (Sartorius Stedim Biotech) and concentrated to desired concentrations and stored at −20 °C until use. All formulations were tested for particle size, distribution, RNA concentration, and encapsulation.

### Flow cytometry measurements
Cells were trypsinized 3 days after transfection and centrifuged at 300 g for 5 min at 4 °C. The supernatant was removed, and the cells were resuspended in $1 \times$ PBS free of calcium or magnesium. Fortessa flow analyzer (BD Biosciences) was used for fluorescence-activated flow analysis with the following settings: EYFP fluorescence was measured with an excitation at 514 nm and an emission in the range of 519–568 nm. tagBFP fluorescence was measured with an excitation at 405 nm and an emission in the range of 410–507 nm. mKate fluorescence was measured with an excitation at 543 nm and an emission in

the range of 571–651 nm. mScarlet fluorescence was measured with an excitation at 488 nm and an emission in the range of 571–651 nm. For each sample, ~1 × $10^5$ to ~5 × $10^5$ cell events were collected. Scatter plots and gating are shown in Supplementary Fig. 10.

## Luciferase assays

Cell culture media were removed 24 h after transfection, and 100 µL of Bright-Light Luciferase Assay System (Vazyme) was added to each well, mixed and incubated for 3 min. 100 µL of the culture medium mixture was transferred to a flat-bottom white-walled plate (Corning) and the luminescence was measured on a microplate luminometer (Thermo Fisher Scientific).

## Lactate dehydrogenase cytotoxicity assays and quantification

Cell death was evaluated using the Lactate Dehydrogenase Cytotoxicity Assay Kit (Beyotime, Shanghai, China). The LDH analysis was performed according to the manufacturer's instructions. Briefly, $1 × 10^5$ cells were seeded into a 96-well culture plate 1 day before transfection, and 0.05 ug (for Fig. 4g) or 0.1 ug (for Fig. 4i) LNP encapsulated circRNAs were transfected into each well next day. Twenty-four hours later, the cell culture plates were centrifuged at 400 g for 5 min and the supernatant was carefully aspirated, and 150 µL of the Lactate Dehydrogenase (LDH) release reagent, diluted ten-fold with Phosphate-Buffered Saline (PBS) (a volumetric ratio of 1:10 of reagent to PBS), was added and mixed thoroughly. After 1-h incubation, the cell culture plate was centrifuged again at 400 g for 5 min and the resultant supernatant (120 µL from each well) was then transferred into corresponding wells of a fresh 96-well plate for subsequent sample quantification. For the quantification of LDH activity in the 96-well plate samples, 60 ul of LDH detection working solution was added to each well (sample volume: detection working solution volume = 2:1). The plate was mixed well and incubated at room temperature (about 25 °C) in the dark for 30 min (the plate was wrapped with aluminum foil and placed on a horizontal shaker or a side-swing shaker for slow shaking). The absorbance was measured at 490 nm, using 655 nm as the reference wavelength for dual-wavelength measurement. The absorbance of the background blank control wells was subtracted from the absorbance of each group. The cell toxicity or mortality (%) was calculated as [(treated sample absorbance – sample control well absorbance) / (cell maximum enzyme activity absorbance – sample control well absorbance)] * 100%.

## Data analysis

For all studies, the data presented are representatives of three independent biological replicates. For flow cytometry measurement, EYFP or mKate (when fully complementary miRBS was included in the construct) fluorescence protein was used for internal control. For each sample, we calculated the mean tagBFP fluorescence intensity value (MFI of tagBFP) and MFI of EYFP or mKate of positive-gated cells. The relative IRES-mediated tagBFP fluorescence intensity was defined as the MFI ratio of tagBFP/EYFP (or tagBFP/mKate when fully complementary miRBS was included in the construct). The normalized IRES-mediated tagBFP fluorescence intensity was defined as the relative IRES-mediated tagBFP fluorescence intensity normalized to a positive control (nIRES or rIRES) tested in the same cell line. Statistical analysis of the results was performed by a two-tailed unpaired Welch's t-test, assuming unequal variances. Differences were considered significant when $p < 0.05$.

## Statistics and reproducibility

Statistical analysis of the results was performed by a two-tailed unpaired Welch's t-test, assuming unequal variances. Differences were considered significant when $p < 0.05$. No special sample size calculations were performed. By convention, experiments were performed in independent biological triplicate (n = 3). No data was excluded from

the analysis. The activity of the IRES reporter was measured in three biological replicates. All attempts at replication were successful. Randomization is not relevant to our study, as all experiments were done in cell culture. Samples were separated based on transfection conditions. Blinding is not applicable in our study, as all experiments were done in cell culture.

## Reporting summary

Further information on research design is available in the Nature Portfolio Reporting Summary linked to this article.

## Data availability

All data supporting the findings of this study are available within the manuscript, Supplementary Information file and Supplementary Data files. The sequence information has been provided in Supplementary Data 1. The raw data generated in this study have been provided within the Source Data file. The annotated sequences have been deposited at the Benchling website under the link https://benchling.com/hui_ning/f_/K81GiYaI-rational-design-of-microrna-responsive-switch-for-programmable-translational-control-in-mammalian-cells/. Source data are provided with this paper.

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

## Acknowledgements

We thank members of Xie lab for helpful discussions. We appreciate the technical supports provided by Beijing Syngentech Co., Ltd. This research is supported by the National Natural Science Foundation of China (No. 61721003 [Z.X.]), National Key Research and Development Program of China (No. 2019YFA0906103 [Z.X.], 2021YFC2302401 [Z.X.]), Beijing Tsinghua Industrial R&D Institute (No. 20222000470 [Z.X.]), Tsinghua University Spring Breeze Fund, and Beijing National Research Center for Information Science and Technology.

## Author contributions

Z.X., H.N., and G.L. conceived of the ideas implemented in this project. H.N. constructed the plasmids and performed plasmid transfection experiments. H.N., G.L., L.L., Q.L., and H.H. synthesized and purified circRNAs, and performed circRNA transfection experiments. Z.X., H.N., and G.L. analyzed the data. Z.X. supervised the project. Z.X., H.N., and G.L. wrote the paper.

## Competing interests

Z.X., H.N., G.L., Q.L., and H.H. as the inventors have filed a patent application to the State Intellectual Property Office of China based on the presented work [PCT/CN2023/112676]. Tsinghua University and Beijing Syngentech Co., Ltd. are the patent applicants. The remaining authors declare no competing interests.
