## [Peer Review File · Nature Communications]

REVIEWER COMMENTS

Reviewer #1 (Remarks to the Author):

Ning et al reported an approach to detect intracellular miRNA with a designed IRES-mediated RNA system. They first engineered HCV IRES, a well-studied IRES, to confer the ability to respond to miRNAs. In the absence of miRNA, the active IRES structure is disrupted, while in the presence of miRNA, miRNA binding alters the IRES structure to activate the IRES-dependent translation. Using this strategy, the authors detected endogenous miRNAs in HEK293 and Huh7 and constructed some logic circuits. Importantly, their strategy could be applicable to circRNA as they demonstrated in Figure 2. Artificial regulation of translation by intracellular RNA is a hot topic in RNA synthetic biology and RNA-based therapeutics and this work is an interesting attempt to expand the toolbox of these fields. However, the data the authors showed in the manuscript is insufficient to support their concept and claims, and I do not recommend publishing it in Nature Communications. I think that numerous additional data and evidence should be required to publish this work.

Major points

1. All statistics (mean values and error bars) the authors showed is derived from three technical replicates. However, statistics should be derived from three (or more) independent experiments unless there is a clear scientific justification.
2. The translation ability of IRES usually depends on the cell types. Therefore, the authors should show the absolute expression levels of the reporter from IRESs in each cell type.

In distinguishing different cell types by using cell-type classifier circuits, the absolute expression level of the output (e.g. fluorescent protein) in each cell type should be compared. If the absolute output expression levels in HEK293 and Huh7 are comparable, these two cell lines cannot be distinguished even if the MITA responds to miRNAs and works as intended. In Figure 1e, i, Figure 2b, and c, the mean tagBFP fluorescence intensities were normalized to rIRES or nIRES in each cell type and, indeed, it seems that Huh7 showed higher reporter expression than HEK293. However, I'm wondering whether it is suitable to eliminate the possible differences in IRES-mediated translation in these cell lines by the normalization and compare the normalized values. Again, information on the absolute reporter expression levels would be helpful.

3. Related to the comment 2, Huh7 is the cell that is derived from the liver and might be compatible with HCV IRES (and HCV-like IRESs). The authors should test IRESs and the PROMITAR platform in other cell types (HEK293 and non-Huh7).
4. The authors normalized the data with rIRES or nIRES in most experiments. However, it is unclear how each miRNA affects the translation from IRES itself. Have the authors verified this point? For example, in Figure 1d, the authors should test co-transfection of control mimic or FF4 mimic with rIRES and dIRES. This point should also be addressed in other experiments.

5. Similar to the comment 4, the response to miRNA should be confirmed with MITA with non-target miRBS too.
6. In some situations, the RNA secondary structure is altered by the intracellular context. There is the possibility that the structural changes of designed IRES might be induced independent of miRNA. For example, in Figure 1e, the authors should show the expression levels in each construct including MITA with non-target miRBS. This point should also be addressed in other experiments.
7. Related to the comment 6, the authors designed a variety of engineered IRESs in Figure 1e. However, only TL16, SL3, and AJ2 were tested in Figure 1d. I think it is important to show generality of design principle of MITAs. Did the authors design and test a variety of FF4-responsive MITA? Is there a suitable design of MITAs for a variety of miRNAs?
8. The experiments that detect endogenous miRNA with MITA are insufficient to support the authors' claims. The authors should use miRNA inhibitors in Huh7 cells to confirm whether increase in the reporter expression is due to endogenous miR-21 activity. In addition, the authors should use the miR-21 mimic in HEK293.
9. Related to Figures 1f and g, it should be confirmed whether the order of miRBS affects the circuit performances.
10. On page 4, line 162, the authors mentioned that miR-199a-3p is a miRNA that shows little expression in HEK293 with reference to Supplementary Figure 2. However, data of miR-199a-3p is not shown in Supplementary Figure 2.
11. Related to Figure 1h, the authors claimed that co-transfection of miRNA-FF4 significantly inhibited the translation of tagBFP on page 4, line 174. However, they did not use a statistical test to test if there is a statistically significant difference between two conditions.
12. Although the authors mentioned on page 5, line 196, the example of application should be demonstrated in the manuscript.
13. Related to the claim mentioned on page 5, line 205, the authors do not provide any data to support that the MITA and MITR strategies could be applicable to other IRESs. The authors should demonstrate the MITA and MITR with these IRESs.
14. What % of cells are converted to reporter-positive cells by miRNA? The authors should show the scatter plots of flow cytometry data. Fluorescence microscopy images will also be helpful.
15. The time scale of experiments is different between plasmid experiments and circRNA experiments. Could the authors explain the reason? Especially, in the plasmid experiments, the authors carried out flow cytometry measurements 3 days after transfection, but I feel it is a little bit longer than usual. Does the performance of each experiment change depend on the time?
16. The supplementary Table 1 is not helpful for readers. The authors should highlight each element by case, bold fonts, italic, underlining, and color, for example.
17. There are some other reports that can detect miRNAs.

(<https://doi.org/10.1101/2022.05.10.491309>, <https://doi.org/10.1021/acssynbio.8b00530>)

These also seem to work in circRNAs in principle. Although these works do not perform circRNA experiments, it is better to discuss the advantages of the authors' work compared with other systems.

Minor points

1. The authors should show the detail of transfection. (e.g. DNA/RNA amount and combinations)
2. In circRNA constructs, why EGFP was included?
3. On page 6, line 232, what are Hela cells used for?
4. On page 6, line 236, the author used Lipofectamine 8000, but is it correct?
5. About reference 18, isn't it requotation

Reviewer #2 (Remarks to the Author):

This paper describes the development of a platform based on a programmable miRNA-responsive IRES for miRNA-based translation control. The authors provide a logical description of their rational design-based approach for engineering the IRES using a well-characterized IRES from the Hepatitis C virus and developing the platform, which is subsequently used use the construct logic gates and cell-type classifiers.

The experiments are well described, and the conclusions are supported by the results presented. The main issue of this study is the lack of evidence supporting the main claim of the paper that the system developed can be programmed to detect any input miRNA. In addition, the manuscript requires significant editing to improve readability and correct syntax/grammar mistakes.

Specific comments:

Line 101: In the discussion of previously reported mi-RNA responsive switches, add Wang et al. 2019, doi: 10.1021/acssynbio.8b00530.

Line 103: What do the authors mean by "not applicable to circRNAs"?

Line 116: What are the four domains and their function? Are there domains other than IV that are targetable?

Line 145: How does the amount of exogenous miRNA provided compare with endogenous levels of miRNA? Was it provided at a comparable amount as endogenous miRNA?

Line 155: Define miRBS

Figure 1c: This figure is too detailed and hard to understand, it would be great if the authors could modify it to emphasize how the three designs differ.

Line 157 / Figure 1e: It seems like some MITAs in HEK293 cells have a basal level of mean tagBFP fluorescence instead of having a near-zero value. Is it an issue of the endogenous miRNA levels in the HEK293 cells or is it leakiness of the sensor design?

Line 177: Add a reference to the figure.

Line 180: the authors do not include experiments and results demonstrating that MITA can be programmed to respond to any miRNA

It would be important to demonstrate the detection of another endogenous miRNA to test the modularity of the PROMITAR system and validate the use of PROMITAR as a platform technology. It is stated that the platform can be used to detect any input miRNA, but no evidence is provided to support this claim.

REVIEWER COMMENTS

Reviewer #1 (Remarks to the Author):

Ning et al reported an approach to detect intracellular miRNA with a designed IRES-mediated RNA system. They first engineered HCV IRES, a well-studied IRES, to confer the ability to respond to miRNAs. In the absence of miRNA, the active IRES structure is disrupted, while in the presence of miRNA, miRNA binding alters the IRES structure to activate the IRES-dependent translation. Using this strategy, the authors detected endogenous miRNAs in HEK293 and Huh7 and constructed some logic circuits. Importantly, their strategy could be applicable to circRNA as they demonstrated in Figure 2. Artificial regulation of translation by intracellular RNA is a hot topic in RNA synthetic biology and RNA-based therapeutics and this work is an interesting attempt to expand the toolbox of these fields. However, the data the authors showed in the manuscript is insufficient to support their concept and claims, and I do not recommend publishing it in Nature Communications. I think that numerous additional data and evidence should be required to publish this work.

Major points

1. All statistics (mean values and error bars) the authors showed is derived from three technical replicates. However, statistics should be derived from three (or more) independent experiments unless there is a clear scientific justification.

We appreciate your careful reading and comments. We apologize for the lack of clarity in our previous manuscript regarding the terminology for independent biological replicates. The data presented in our previous manuscript were not "technical replicates" but "independent biological replicates", as we stated in the previous Reporting Summary and the updated Reporting Summary files. To further validate the reliability and reproducibility of our results, we have performed additional experiments and updated our data accordingly. The additional experimental data and all subsequent supplementary experiments conducted have been stated as being the result of three independent biological replicates.

2. The translation ability of IRES usually depends on the cell types. Therefore, the authors should show the absolute expression levels of the reporter from IRESs in each cell type.

In distinguishing different cell types by using cell-type classifier circuits, the absolute expression level of the output (e.g. fluorescent protein) in each cell type should be compared. If the absolute output expression levels in HEK293 and Huh7 are comparable, these two cell lines cannot be distinguished even if the MITA responds to miRNAs and works as intended. In Figure 1e, i, Figure 2b, and c, the mean tagBFP fluorescence intensities were normalized to rIRES or nIRES in each cell type and, indeed, it seems that Huh7 showed higher reporter expression than HEK293. However, I'm wondering whether it is suitable to eliminate the possible differences in IRES-mediated translation in these cell lines by the normalization and compare the normalized values. Again, information on the absolute reporter

expression levels would be helpful.

We appreciate your suggestion and have provided the absolute expression data for all our experiments in this study, including raw data of mean fluorescence intensity for fluorescent protein reporters and raw data of luminescence intensity characterizing luciferase activities. As you pointed out, the basal expression of IRES does differ between cell lines. We found that in plasmid transfection (Supplementary Fig. 1f, Fig. 1g), and in transfection with in vitro transcribed circular RNAs (Supplementary Fig. 5a), the basal expression of HCV IRES in Huh7 is approximately 1.2 - 1.5 and 2-3 times higher than that in HEK293, respectively. Therefore, when considering the differences in the basal expression level of IRES and the higher normalized expression of MITA and cell-type classifier in Huh7 compared to HEK293, it leads to a much higher absolute expression of MITA and cell-type classifier in Huh7 (Supplementary Fig. 4b, 4j, Supplementary Fig. 5b, 5c), which can be applied to cell classification and related applications, such as targeting and killing of tumor cells (Fig.3g, Fig.3i).

3. Related to the comment 2, Huh7 is the cell that is derived from the liver and might be compatible with HCV IRES (and HCV-like IRESs). The authors should test IRESs and the PROMITAR platform in other cell types (HEK293 and non-Huh7).

We appreciate your suggestion to test the performance of MITAs in other cell lines. We have now tested the performance of miR-21 responsive MITAs in HeLa cell lines (another cancer cell line with high miR-21 activity). The results showed that the reporter tagBFP expression of miR-21 responsive MITAs in HeLa cells (miR-21 high activity) was also higher than that in HEK293 cells (miR-21 low activity), with a similar trend as we observed with Huh7, which indicated that the engineered HCV IRES could function in other cell lines (Supplementary Fig. 4f).

4. The authors normalized the data with rIRES or nIRES in most experiments. However, it is unclear how each miRNA affects the translation from IRES itself. Have the authors verified this point? For example, in Figure 1d, the authors should test co-transfection of control mimic or FF4 mimic with rIRES and dIRES. This point should also be addressed in other experiments.

We appreciate your comment about how each miRNA affects the translation from IRES itself. The reason we normalized the expression of engineered IRES with either nIRES or rIRES across different cell lines was to eliminate inherent expression differences of IRES in different cell lines (as mentioned above at major point 2), allowing us to focus on the single impact of our engineering strategies on IRES translational activity. Following your suggestion, we have now added control experiments to investigate the potential effects of miRNA on the expression of nIRES, dIRES, and rIRES (Supplementary Fig. 2g). Our results showed that control mimics, exogenous miR-FF4 mimics, and endogenous miR-21 mimics did not affect the basal expression level of nIRES or rIRES.

5. Similar to the comment 4, the response to miRNA should be confirmed with MITA with non-target miRBS too.

We appreciate your suggestion to validate the specificity of miR-FF4 responsive MITA. To investigate the specificity of our miR-FF4 responsive MITA, we have added control mimics and non-target miR-199a mimics as non-target miRNA to evaluate whether the miR-FF4 responsive MITA was responsive to the non-target miRNA (Supplementary Fig. 2h). The results of these control experiments indicated that the miR-FF4 responsive MITA did not respond to non-target miRNA (i.e., miR-199a).

6. In some situations, the RNA secondary structure is altered by the intracellular context. There is the possibility that the structural changes of designed IRES might be induced independent of miRNA. For example, in Figure 1e, the authors should show the expression levels in each construct including MITA with non-target miRBS. This point should also be addressed in other experiments.

We appreciate your suggestion to further test the specificity of our MITA design. To test whether the structural changes of the designed IRES were induced by intracellular context instead of the target miRNAs, we transfected different plasmid constructs of exogenous miR-FF4 responsive MITA into HEK293 and Huh7 cells without co-transfecting miR-FF4 mimics (Supplementary Fig. 4c). The results showed that the expression difference of MITA containing non-target miRBS (i.e., miR-FF4 binding sites) in HEK293 and Huh7 did not exceed that of IRES itself between these two cell lines. While the transfection experiments of MITA containing miR-21 binding sites showed higher expression difference between HEK293 and Huh7 (Fig. 2g). Taken together, these results ruled out the possibility that the structural changes of designed IRES might be induced by the intracellular context instead of target miRNA.

7. Related to the comment 6, the authors designed a variety of engineered IRESs in Figure 1e. However, only TL16, SL3, and AJ2 were tested in Figure 1d. I think it is important to show generality of design principle of MITAs. Did the authors design and test a variety of FF4-responsive MITA? Is there a suitable design of MITAs for a variety of miRNAs?

We appreciate your suggestion to demonstrate the generality of the design principle of MITAs. We have expanded our experiments to include a variety of exogenous miR-FF4 responsive MITAs (including various designs of TL14 to TL18, SL0 to SL6, and AJ0 to AJ5, Fig. 1j, Fig. 1k, Fig. 1l), a variety of endogenous miR-18a responsive MITAs (including various design of TL14 to TL18, SL0 to SL6, and AJ0 to AJ5, Fig. 2h), and a variety of endogenous miR-21 responsive MITAs based on CSFV IRES (Fig. 4e). Taken together with the results of miR-21 responsive MITAs (Fig. 2g), We concluded that TL16, SL3, and AJ2 are generally suitable designs of MITA for a variety of miRNAs.

8. The experiments that detect endogenous miRNA with MITA are insufficient to support the authors' claims. The authors should use miRNA inhibitors in Huh7 cells to confirm whether increase in the

reporter expression is due to endogenous miR-21 activity. In addition, the authors should use the miR-21 mimic in HEK293.

We appreciate your careful reading and helpful suggestion. We have tested miR-21 responsive MITAs in Huh7 cells with miR-21 inhibitor co-transfection and in HEK293 with miR-21 mimics co-transfection (Supplementary Fig. 4d, 4e). The results showed that the reporter tagBFP expression decreased with miR-21 inhibitor co-transfection in Huh7, and increased with miR-21 mimic co-transfection in HEK293. These results confirmed that these MITA designs were indeed responsive to miR-21. These results should confirm the increase in the IRES reporter expression was due to endogenous miR-21 activity.

9. Related to Figures 1f and g, it should be confirmed whether the order of miRBS affects the circuit performances.

We appreciate your suggestion about testing the effect of the order of miRBS on OR-gates and AND-gates. We swapped the order of miRBS in OR-gates and AND-gates design, and the performance of these logic-gate design was consistent with our previous findings, indicating that our logic-gate design were not affected by the arrangement of miRBS (Supplementary Fig. 3g, 3h).

10. On page 4, line 162, the authors mentioned that miR-199a-3p is a miRNA that shows little expression in HEK293 with reference to Supplementary Figure 2. However, data of miR-199a-3p is not shown in Supplementary Figure 2.

We thank you for careful reading and pointing out the missing data on miR-199a-3p activity in HEK293. We apologize for the oversight of miR-199a-3p data in HEK293 in our previous manuscript. We have now included this data in Supplementary Fig. 3a. The data showed little miRNA activity of miR-199a-3p in HEK293 cells.

11. Related to Figure 1h, the authors claimed that co-transfection of miRNA-FF4 significantly inhibited the translation of tagBFP on page 4, line 174. However, they did not use a statistical test to test if there is a statistically significant difference between two conditions.

Thank you for pointing out the lack of statistical testing in our manuscript. We have now included a two-tailed unpaired Welch's t-test in our data (Fig. 2f), which confirms the statistical significance of our results ($p = 0.0029$).

12. Although the authors mentioned on page 5, line 196, the example of application should be demonstrated in the manuscript.

We thank you for suggesting the demonstration of the application of our PROMITAR platform. We have performed the lactate dehydrogenase (LDH) cytotoxicity assay to evaluate the specificity of our engineered circRNAs for tumor cell targeting and killing. For the LDH cytotoxicity assay, we replaced the luciferase reporter of circRNAs with the N-terminal section of Gasdermin D (GSDMD), a pore-

forming protein that is involved in the inflammatory cell death pathway known as pyroptosis (Fig. 3g, 3i). The results showed substantial differential cytotoxicity (up to 18-fold) between Huh7 and HEK293 cells when we transfected AJ2 MITA-containing circRNA (Fig. 3g). Notably, when we transfected the cell-type classifier circRNA into Huh7 and HEK293, the differential cytotoxicity was up to 89-fold (Fig. 3g). These results indicated potential therapeutic applications of our PROMITAR platform for targeted cancer cell killing.

13. Related to the claim mentioned on page 5, line 205, the authors do not provide any data to support that the MITA and MITR strategies could be applicable to other IRESs. The authors should demonstrate the MITA and MITR with these IRESs.

We appreciate your suggestion and have performed additional experiments of engineering CSFV IRES for miR-21 responsive MITAs and the miR-FF4 responsive MITR, respectively. We transfected these MITA-containing reporter plasmids into Huh7 and HEK293 cells (Fig. 4e), and co-transfected MITR-containing reporter plasmids with miRNA mimics in HEK293 cells (Fig. 4f, 4g). Both the MITA and MITR design based on CSFV IRES produced similar results to the MITA and MITR design based on HCV IRES, which demonstrated the versatility of our design principle.

14. What % of cells are converted to reporter-positive cells by miRNA? The authors should show the scatter plots of flow cytometry data. Fluorescence microscopy images will also be helpful.

We acknowledge your suggestion for showing the percentage of cells converted to reporter-positive cells by miRNA. Our experiments yielded a transfection efficiency range of 60%-80%. To provide further clarity, we have provided scatter plots of flow cytometry data (Fig. 1f, Fig. 4d) and fluorescence microscopy images (Fig. 1e) in the manuscript, and we also provided the scatter plots of flow cytometry data in MITA-testing experiments (Supplementary Fig. 6f), the translation efficiency was in a range of 70%-80% and the percentage of cells was in a range of 60%-70%.

15. The time scale of experiments is different between plasmid experiments and circRNA experiments. Could the authors explain the reason? Especially, in the plasmid experiments, the authors carried out flow cytometry measurements 3 days after transfection, but I feel it is a little bit longer than usual. Does the performance of each experiment change depend on the time?

We thank you for careful reading of our manuscript. In our plasmid transfection experiments, we noted that the IRES-initiated tagBFP translation was slower than the cap-initiated EYFP translation. The tagBFP protein expression of IRES-initiated translation has not yet achieved the highest level at 48 hours. Therefore, we decided to carry out flow cytometry measurements 3 days after plasmid transfection.

16. The supplementary Table 1 is not helpful for readers. The authors should highlight each element by case, bold fonts, italic, underlining, and color, for example.

We have now annotated each element by case and formatted it for readability. The annotated plasmid

sequences used in this study are now deposited at the Benchling websites:

https://benchling.com/hui_ning/f_/K81GiYaI-rational-design-of-microrna-responsive-switch-for-programmable-translational-control-in-mammalian-cells/

17. There are some other reports that can detect miRNAs.

(<https://doi.org/10.1101/2022.05.10.491309>, <https://doi.org/10.1021/acssynbio.8b00530>)

These also seem to work in circRNAs in principle. Although these works do not perform circRNA experiments, it is better to discuss the advantages of the authors' work compared with other systems.

We thank you for highlighting other reports on miRNA detection. We have discussed these in our revised manuscript and explained the advantage of our approach over these existing systems. Briefly, both systems mentioned cannot be effectively applied to circular RNAs for the following reasons. The bioRxiv paper's strategy depends on miRNA cleavage to eliminate the translation suppression sequence, but any cleavage on circRNAs would break down circRNAs and result in degradation of circRNAs. The ACS Synthetic Biology paper's strategy relies on regulating the cap-initiated ribosome scanning and targeting by controlling the local RNA secondary structure of Kozak sequence. However, as the translation initiation of circRNA is cap-independent, this strategy might not be feasible in circRNAs either.

Minor points

1. The authors should show the detail of transfection. (e.g. DNA/RNA amount and combinations)

We thank you for your careful reading and helpful comments. We have provided details of transfection including DNA and RNA amounts in the Methods section.

2. In circRNA constructs, why EGFP was included?

We thank you for your careful reading. EGFP was included in our circRNA constructs as it is part of the original plasmid backbone used in our lab for verifying successful RNA transfection. However, for a more precise quantification of the translation level of circRNAs, we chose to measure the luminescence signal for luciferase activity.

3. On page 6, line 232, what are HeLa cells used for?

We apologize for our oversight in not explaining the use of HeLa cells in our initial manuscript. HeLa cells were used as an additional cell line for testing our MITA design. We have now included related data in our revised manuscript (Supplementary Fig. 4f).

4. On page 6, line 236, the author used Lipofectamine 8000, but is it correct?

We thank you for your careful reading. The Lipofectamine 8000 mentioned in our manuscript is a transfection reagent from the company named Beyotime. Lipofectamine 8000 (Beyotime) is different from the typical transfection reagent Lipofectamine 3000 (ThermoFisher).

5. About reference 18, isn't it requotation

We thank you for your careful reading. We have double-checked our references and found that reference 18 is not requotation.

Reviewer #2 (Remarks to the Author):

This paper describes the development of a platform based on a programmable miRNA-responsive IRES for miRNA-based translation control. The authors provide a logical description of their rational design-based approach for engineering the IRES using a well-characterized IRES from the Hepatitis C virus and developing the platform, which is subsequently used use the construct logic gates and cell-type classifiers.

The experiments are well described, and the conclusions are supported by the results presented. The main issue of this study is the lack of evidence supporting the main claim of the paper that the system developed can be programmed to detect any input miRNA. In addition, the manuscript requires significant editing to improve readability and correct syntax/grammar mistakes.

Specific comments:

Line 101: In the discussion of previously reported mi-RNA responsive switches, add Wang et al. 2019, doi: 10.1021/acssynbio.8b00530.

We appreciate your suggestion and the careful reading of our manuscript. We have included the reference of Wang et al. 2019 published on ACS Synthetic Biology, and discussed the distinction between our work and the cited article in the revised manuscript. The strategy employed in the study of Wang et al. relies on blocking the Kozak sequence and hindering the cap-initiated ribosome scanning to turn off the translation, while the binding of target miRNA enables the release of the Kozak sequence and the cap-initiated scanning ribosome could recognize and bind to the Kozak sequence to turn on the translation. However, as the translation initiation in circular RNAs (circRNAs) is cap-independent, this strategy might not be feasible in circRNAs.

Line 103: What do the authors mean by “not applicable to circRNAs”?

We thank you for pointing out the clarification needed for the applicability of previously reported cleavage-based miRNA-ON switches in circRNAs. By stating “not applicable in circRNA”, we meant that the previous strategies required RNA cleavage, but any cleavage on circRNAs would break down circRNAs and result in degradation of circRNAs, rendering these strategies not applicable to circRNAs.

Line 116: What are the four domains and their function? Are there domains other than IV that are

targetable?

We appreciate your request for an explanation of the Hepatitis C virus (HCV) IRES's four domains and their roles. Briefly, domain I of the HCV IRES is a stem-loop structure that is located at the 5' end of the HCV genome. Domain II primarily interacts with the ribosomal 40S subunit. Domain III is required for 40S recruitment and interacts with eIF3. Domain IV contains a pseudoknot structure and the AUG start codon, both of which are essential for translation initiation. We designed the experiment targeting the domain IV at first because of its importance in translation and the pseudoknot is sterically adjacent to the 5' end of HCV IRES, which we thought were essential factors for RNA base-pairing and translation interference. Since the design of domain IV targeting worked well and we then focus only on targeting this domain and designed further sophisticated IRES translation modulation structures.

Line 145: How does the amount of exogenous miRNA provided compare with endogenous levels of miRNA? Was it provided at a comparable amount as endogenous miRNA?

Thank you for pointing out the comparison between exogenous and endogenous miRNA levels. We agree that the target miRNA concentration is a critical factor in our system's operation. We transfected the miRNA mimics with a final transfection concentration of 50 nM and the concentrations of transfected miRNA mimics are typically higher than the endogenous miRNAs. In addition, we supplemented dosage-response experiments in our studies involving exogenous miR-FF4 responsive MITA (Fig. 1j, 1k, 1l) and found that our devices exhibit dose-dependent activity.

Line 155: Define miRBS

Thank you for suggesting the need to define miRBS. We have now included the definition of miRBS (miRNA binding site) in the main text, and we have also added definitions for the abbreviations in figure legends.

Figure 1c: This figure is too detailed and hard to understand, it would be great if the authors could modify it to emphasize how the three designs differ.

We appreciate your suggestion to improve Fig. 1c of our previous manuscript. The three types of MITA design mainly differ in the local secondary structure sequestering the rescue sequence (RS). Following your suggestion, we have divided this figure according to the three types of design (toehold-like, stem-loop, 3-arm-junction) and presented them as three separate figures (Fig. 1g, 1h, and 1i) in the revised manuscript to enhance clarity.

Line 157 / Figure 1c: It seems like some MITAs in HEK293 cells have a basal level of mean tagBFP fluorescence instead of having a near-zero value. Is it an issue of the endogenous miRNA levels in the HEK293 cells or is it leakiness of the sensor design?

We appreciate your careful reading and pointing out the non-zero basal expression of miR-21 designed

MITA in HEK293 cells. The non-zero basal expression of the miR-21 designed MITA in HEK293 can be attributed to both reason: the expression of endogenous miR-21 in HEK293 (refer to Supplementary Fig. 3a, where the activity of miR-21 in HEK293 is not zero) and the leakiness of the designed device (refer to Fig1j, 1k, 1l; even with the miR-FF4 responsive MITAs, there is a non-zero basal expression when control mimics are added). We think the leakiness is primarily due to the flexibility of RNA structures, which means that some RNA molecules may not folded as our designed structure. These non-folded RNA molecules may contribute to the leakiness.

Line 177: Add a reference to the figure.

We appreciate your suggestion and we have now added the reference to the figure you mentioned. Additionally, the data presented in this figure were results produced by our own experiments in this manuscript.

Line 180: the authors do not include experiments and results demonstrating that MITA can be programmed to respond to any miRNA.

Thank you for highlighting the need for evidence to support the broad applicability of our MITA design. In the previous manuscript, we designed MITA to detect endogenous miRNA-21 and exogenous miRNA-FF4 signals. To further demonstrate the versatility of our MITA design, we have tested MITA with another endogenous miRNA (miR-18a, see Fig. 2h) which has high miRNA activity in HEK293 but low activity in Huh7 cells (see Supplementary Fig. 3a, 3b). The results showed that the tagBFP reporter expression was higher in HEK293 cells than that in Huh7 cells. The miRNA activities of miR-18a and miR-21 in HEK293 and Huh7 cell lines were opposite, and the differential expression patterns of the miR-18a and miR-21 responsive MITAs in HEK293 and Huh7 were also inversely proportional. Taken together, these results further validated the broad applicability of our MITA design for other miRNA signal detections.

It would be important to demonstrate the detection of another endogenous miRNA to test the modularity of the PROMITAR system and validate the use of PROMITAR as a platform technology. It is stated that the platform can be used to detect any input miRNA, but no evidence is provided to support this claim.

REVIEWER COMMENTS

Reviewer #1 (Remarks to the Author):

The manuscript revised by Ning et al. has been improved through numerous additional experiments and revisions. However, there are still several concerns and a considerable number of errors present in the manuscript.

The major concern is that the use of the IRES itself can lead to cell type-dependent differences in basal translation levels, limiting the broad applicability of this system. Fortunately, most data indicate that this system functions in a miRNA-dependent manner. However, since the IRES itself can classify cell types, there is no longer a need to introduce miRNA dependence. Furthermore, in real-world applications, there will likely be a need to identify IRESs that have high translational levels in the target cells and use PROMITAR accordingly.

The particular concerns are as follows:

1. The authors successfully designed a system with improved performance by effectively harnessing the higher translation activity of IRES in Huh7 cells (Fig 3f and i). However, it is unclear whether this is a reflection of miRNA activity or a specific capability of the modified IRES itself. Furthermore, it would be desirable to discuss the reasons behind the increased translational activity of IRES with miRNA-responsiveness compared to native IRES (or engineered IRES with the same structure).

2. It remains unclear how far the applicability extends, as the observed higher efficacy is primarily seen in cells with high translation activity. Conversely, is it possible to selectively induce cell death in only 293FT cells? In particular, it seems that 293FT cells are less susceptible to cell death in this assay system, while Huh7 cells seem to be more prone to cell killing. Considering this point, the observed results may be contributed by the synergistic effect of the variation in IRES activity among different cell types, the susceptibility to cell death exhibited by different cell types, and MITA due to miRNA activity. Therefore, the importance of miRNA-responsiveness contribution remains uncertain. Related to this point, it would be desirable to conduct co-cultures of different cell types to determine if the target cells can be isolated (or distinguished) from others. Performing experiments with multiple cell types would be much preferable.

3. No experimental evidence has been provided to demonstrate the advantages gained by incorporating PROMITAR into circRNA. What specific functionalities can only be achieved with miRNA-responsive circRNA that cannot be accomplished with linear mRNA?

Major points

4. Some bar graphs include individual data points. This representation method should be employed for all data.

5. Lines 198-199. Related to Figure 2j, the authors described that they observed a 3-fold increase of fluorescent signal when they employed MITA or MITR. I understood that the nIRES (with only inhibiting miRBS) is a condition employing only MITR and showed an approximately 2-fold increase in fluorescent signal in Huh7 than HEK293. However, which is a condition where only MITA was employed? If only MITR was employed in this experiment, please revise the manuscript.

6. Materials and methods section. Please describe the protocol of measurement of LDH assay. The current manuscript only describes a protocol for sample preparation for LDH assay but lacks a protocol for quantitative measurement.

7. There seem to be errors in explanations in Supplementary Figures 4 and 6.

- In panel A of Supplementary Figure 4, “Mean fluorescence intensity data related to Fig. 2d (HEK293)” seems to be “Mean fluorescence intensity data related to Fig. 2g (HEK293).” Also, “Fig. 2d (Huh7)” seems to be “Fig. 2g (Huh7).”

- In panel G of Supplementary Figure 4, “Fig. 2e (HEK293)” seems to be “Fig. 2h (HEK293).” Also, “Fig. 2e (Huh7)” seems to be “Fig. 2h (Huh7).”

- In panel I of Supplementary Figure 4, “Fig. 2f (HEK293)” seems to be “Fig. 2j (HEK293).” Also, “Fig. 2f (Huh7)” seems to be “Fig. 2j (Huh7).”

- In panel D of Supplementary Figure 6, “Mean fluorescence intensity data related to Fig. 4d (HEK293)” seems to be “Mean fluorescence intensity data related to Fig. 4e (HEK293).” Also, “Fig. 4d (Huh7)” seems to be “Fig. 4e (Huh7).”

8. Overall, it is unclear about usefulness of circular-ON switch described in this study due to sensitivity and cell-type-dependent IRES activity.

Minor points

1. There are several grammatical errors in the manuscript. For example, “in presence/absence of” and “To further explored our MITAs.” The manuscript would benefit from language editing by a native English speaker.

2. Line 83: Please correct “human embryo kidney” to “human embryonic kidney.”

3. There are orthographic variants in miR-199a (e.g., miR199a, miRNA-199a-3p, miR-199a-3p). Please use consistent terminology.

4. Please correct “in-vitro” to “in vitro.”

5. Related to the previous minor point 4, the authors should carefully revise the manuscript. I think “Lipo8000” is the correct material rather than “Lipofectamine 8000”.

Reviewer #2 (Remarks to the Author):

the authors addressed the reviewer's comments

REVIEWER COMMENTS

Reviewer #1 (Remarks to the Author):

The manuscript revised by Ning et al. has been improved through numerous additional experiments and revisions. However, there are still several concerns and a considerable number of errors present in the manuscript.

The major concern is that the use of the IRES itself can lead to cell type-dependent differences in basal translation levels, limiting the broad applicability of this system. Fortunately, most data indicate that this system functions in a miRNA-dependent manner. However, since the IRES itself can classify cell types, there is no longer a need to introduce miRNA dependence. Furthermore, in real-world applications, there will likely be a need to identify IRESs that have high translational levels in the target cells and use PROMITAR accordingly.

We sincerely appreciate the reviewer taking the time and effort to thoroughly review our revised manuscript and provide insightful feedback that helped improve the quality of our work. We understand the reviewer's major concern raised regarding whether intrinsic cell-type specific IRES activity might limit the broad applicability and necessity of introducing miRNA regulation. However, we think that our PROMITAR system still has significant advantages and a broad applicability. There are several key advantages provided by introducing miRNA-based regulation through our PROMITAR system.

First, we agree that identifying IRESs with desired inherent expression biases might be a potential approach to construct cell-type classifiers. However, such an approach may lack a rational design strategy, would not allow rational design of customizable cell classifiers or logic functions. IRESs identified in this manner would be confined to specific cell lines in which they exhibit high activity and would be unsuitable for customized cell-type classification or logic computation functions. In contrast, our PROMITAR system employs a rational design strategy, incorporating miRNA-responsive ON and OFF switches to introduce cell-type specificity independent of intrinsic IRES activity. This modularity allows us to target different cell types (e.g., Huh7, HEK293, L-02, etc. as demonstrated in our study) by simply modifying miRNA binding sites, obviating the need for the identification of new biased IRESs for each application.

Second, our PROMITAR can selectively amplify cellular responses by magnifying existing small IRES biases into large expression differences between cell types. As we demonstrated that combining miRNA-responsive ON and OFF switches with intrinsic IRES biases leads to a far greater differential expression between cell types compared with the intrinsic IRES biases alone. For instance, introducing miRNA regulation

expanded the intrinsic 5.8-fold IRES difference between Huh7 and HEK293 cells to ~90-fold (**Fig. 3g, 3i**), enabling more effective cell classification and selective killing. This not only enhances cell-type specificity but also has clinical implications by widening the therapeutic window of treatments and reducing the leakage expression in non-target cells.

Third, despite lower HCV IRES activity in HEK293 than Huh7, in the current revised manuscript we also demonstrated the versatility of our system by constructing cell-type classifiers that selectively induce cell death in HEK293 cells while sparing Huh7 cells (**Supplementary Fig. 8**). We also showcased to employ our PROMITAR system to classify L-02 and Huh7 cell lines in the current revised manuscript. Therefore, our approach can create distinct cell-type expression patterns based on cell-type miRNA profiles, independent of intrinsic IRES variations.

Finally, we fully agree that, for real-world applications, it would be ideal to first identify IRESs with intrinsic biases for the target cell types when possible, and then apply PROMITAR to further amplify the selectivity. However, even in the absence of such cell-type biases of IRES activity, our PROMITAR system can still introduce desired cell-type specificity based solely on miRNA profiles, thus providing a versatile and powerful tool for various applications such as logical operations, cell classification, and targeted therapies.

We hope these clarifications adequately address the reviewer's major concern, and thank the reviewer again for insightful comments which have helped strengthen our work. The following are our point-by-point responses to the reviewer's specific concerns.

The particular concerns are as follows:

1. The authors successfully designed a system with improved performance by effectively harnessing the higher translation activity of IRES in Huh7 cells (Fig 3f and i). However, it is unclear whether this is a reflection of miRNA activity or a specific capability of the modified IRES itself. Furthermore, it would be desirable to discuss the reasons behind the increased translational activity of IRES with miRNA-responsiveness compared to native IRES (or engineered IRES with the same structure).

We thank the reviewer for the constructive comments. For the first concern, there are several experiment results demonstrating that the translation activities of our MITA modules are conferred by miRNA rather than inherent capabilities of the modified IRES:

Through experiments with miRNA mimics and inhibitors, we have shown that the translation activities of our MITA designs were indeed responsive to the presence of corresponding miRNAs in different cell lines (**Supplementary Fig. 2h, 4d, 4e**). In the absence of cognate miRNAs, the translation activities of MITAs were inhibited. While in the presence of the cognate miRNAs, the translation activities of MITA were rescued.

These results indicate that the miRNA regulation, rather than intrinsic IRES properties, dictates the translation activities of the MITA modules.

We have also designed multiple MITA structures, including several Toehold-Like/Stem-Loop/3-Arm-Junction for miR-FF4 (**Supplementary Fig. 2b, 2d, 2f, 2h**), miR-18a (**Supplementary Fig. 4h**) and miR-21 (**Supplementary Fig. 4b**). Only the MITA designs with matching miRNA binding sites were responsive to the corresponding miRNAs (i.e., only the miR-21 responsive MITA modules showed translation activities upregulation in Huh7, but not responded to miR-FF4 or miR-18a). These results suggested that, rather than the modified IRES structures themselves, the miRBS sequences were the key determinants conferring translational activity upregulation. Moreover, when we swapped the miRBS sequences but maintained the same complementary pairing structures in MITA and MITR designs, the effects were reversed (**Supplementary Fig. 8**). This further verifies that the miRNA regulation, rather than structural elements, underlies the functionality of our designs.

For the second concern, previous studies have reported that miR-Ago complexes could activate the translation of HCV IRES. This may explain the enhanced translation activities of our engineered MITA modules compared to the native IRES. The following response figure is derived from the reference No. 20 from the main text (Mengardi C, Limousin T, Ricci EP, Soto-Rifo R, Decimo D, Ohlmann T. microRNAs stimulate translation initiation mediated by HCV-like IRESes. *Nucleic Acids Res* 45, 4810-4824 (2017)). Panel A, B and C showed the different design of miRNA seed match sequence in the 5'-end of the HCV IRES. Panel D and E showed that in the presence of cognate miRNA mimics, the translation activity of HCV IRES increased compared with the control group (random miRNA mimics or mutated miRNA mimics), which could be viewed as the native HCV IRES expression level.

response figure

2. It remains unclear how far the applicability extends, as the observed higher efficacy is primarily seen in cells with high translation activity. Conversely, is it possible to selectively induce cell death in only 293FT cells? In particular, it seems that 293FT cells are less susceptible to cell death in this assay system, while Huh7 cells seem to be more prone to cell killing. Considering this point, the observed results may be contributed by the synergistic effect of the variation in IRES activity among different cell types, the susceptibility to cell death exhibited by different cell types, and MITA due to miRNA activity. Therefore, the importance of miRNA-responsiveness contribution remains uncertain. Related to this point, it would be desirable to conduct co-cultures of different cell types to determine if the target cells can be isolated (or distinguished) from others. Performing experiments with multiple cell types would be much preferable.

We appreciate the reviewer raising the important concern regarding the applicability and generalizability of our system. To address this concern, we have conducted additional experiments to demonstrate that our approach can be applied to selectively identify and eliminate other cell types beyond Huh7 cells, and we also conducted the co-cultures of different cell types to further validate our results.

Specifically, we have constructed a new circRNA classifier to selectively induce cell death of HEK293 cells instead of Huh7 cells, using miR-18a MITA modules and miR-21 MITR modules (classifier 2, **Supplementary Fig. 8a**). We measured cell killing efficacy in HEK293 cells using the same LDH cytotoxicity assays, and observed much higher cytotoxicity in HEK293 compared with Huh7 cells.

In addition, we have generated stable EYFP-Huh7 and mScarlet-HEK293 cell lines (**Supplementary Fig. 7**). By co-culturing these two cell lines and transfection of different circRNA classifiers, we evaluated the changes in percentage of fluorescence-positive cells after treatment. We observed that the Huh7-targeting classifiers (classifier 1) decreased the percentage of EYFP⁺ Huh7 cells (**Fig. 3j, 3k**), while the HEK293-targeting classifiers (classifier 2) reduced the percentage of mScarlet⁺ HEK293 cells (**Supplementary Fig. 8e, ef**).

Moreover, we extended the application of PROMITAR to another cell line L-02, which highly expressed let-7c compared with Huh7 cells. With miR-21 as the Huh7-specific marker and let-7c as the L-02-specific marker, we designed two additional classifiers: classifier with miR-21 MITA and let-7c MITR to distinguish Huh7 from L-02 (classifier 3), and classifier with let-7c MITA and miR-21 MITR to distinguish L-02 from Huh7 (classifier 4) (**Supplementary Fig. 8a**). Transfection experiments showed classifier 3 induced stronger Huh7 cell death compared to L-02 (**Supplementary Fig. 8d, 8g, 8h**), while classifier 4 was more cytotoxic to L-02 versus Huh7. Collectively, these results demonstrated that our PROMITAR-based cell-type classifier design principle could distinguish cell types based on their unique miRNA expression profiles

and could be applied broadly and selectively to desired cell types beyond the proof-of-concept demonstrations in Huh7 cells.

3. No experimental evidence has been provided to demonstrate the advantages gained by incorporating PROMITAR into circRNA. What specific functionalities can only be achieved with miRNA-responsive circRNA that cannot be accomplished with linear mRNA?

We appreciate the reviewer raising this point about the advantages of incorporating our system into circRNA. We observed that the ON-OFF ratios of the MITA are higher when we introduced circRNAs rather than plasmids into mammalian cells (please compare **Fig. 2g, 2j** and **Fig. 3c, 3e**), suggesting the advantages gained by incorporating PROMITAR into circRNA. This may be attributed to differences in translation initiation mechanisms between circRNAs and linear mRNAs.

We also agree with the reviewer's comment that it is interesting to explore specific functionalities can only be achieved with miRNA-responsive circRNA that cannot be accomplished with linear mRNA. However, that is beyond the scope of this study. The primary goal of our work is to develop a simple and effective strategy for controlling IRES-initiated translation. The PROMITAR system proposed in this study is applicable to both circRNAs and linear mRNAs. Nevertheless, our results demonstrated that it is possible to engineer an orthogonal and programmable regulation of protein translation by using the PROMITAR system, which may not interfere with the 5'-cap-initiated translation.

Major points

4. Some bar graphs include individual data points. This representation method should be employed for all data.

We appreciate the reviewer for highlighting the inconsistency in the representation of data points in bar graphs. We have now updated all bar graph figures to include individual data points.

5. Lines 198-199. Related to Figure 2j, the authors described that they observed a 3-fold increase of fluorescent signal when they employed MITA or MITR. I understood that the nIRES (with only inhibiting miRBS) is a condition employing only MITR and showed an approximately 2-fold increase in fluorescent signal in Huh7 than HEK293. However, which is a condition where only MITA was employed? If only MITR was employed in this experiment, please revise the manuscript.

We are grateful to the reviewer for the detailed examination of the data presented

in **Figure 2j**. We apologize for the confusion. The condition using only MITR is the nIRES with inhibiting miRBS, as correctly pointed out, which led to the ~2-fold increase in Huh7 vs HEK293 cells. The condition where only MITA was employed was showed in previous **Figure 2g** (TL16, SL3 and AJ2 conditions). We have corrected the text accordingly and the manuscript has been revised to accurately represent the conditions where only MITA or MITR was employed.

6. Materials and methods section. Please describe the protocol of measurement of LDH assay. The current manuscript only describes a protocol for sample preparation for LDH assay but lacks a protocol for quantitative measurement.

We thank the reviewer for pointing out the incomplete description of the LDH assay protocol. The Methods section has been updated to include the detailed description of the quantitative measurement of the LDH assay.

7. There seem to be errors in explanations in Supplementary Figures 4 and 6.

- In panel A of Supplementary Figure 4, “Mean fluorescence intensity data related to Fig. 2d (HEK293)” seems to be “Mean fluorescence intensity data related to Fig. 2g (HEK293).” Also, “Fig. 2d (Huh7)” seems to be “Fig. 2g (Huh7).”

- In panel G of Supplementary Figure 4, “Fig. 2e (HEK293)” seems to be “Fig. 2h (HEK293).” Also, “Fig. 2e (Huh7)” seems to be “Fig. 2h (Huh7).”

- In panel I of Supplementary Figure 4, “Fig. 2f (HEK293)” seems to be “Fig. 2j (HEK293).” Also, “Fig. 2f (Huh7)” seems to be “Fig. 2j (Huh7).”

- In panel D of Supplementary Figure 6, “Mean fluorescence intensity data related to Fig. 4d (HEK293)” seems to be “Mean fluorescence intensity data related to Fig. 4e (HEK293).” Also, “Fig. 4d (Huh7)” seems to be “Fig. 4e (Huh7).”

We appreciate the reviewer catching these errors in in the references to panels in Supplementary Figures 4 and 6. We have double checked and corrected all the wrong figure number references in Supplementary Figures.

8. Overall, it is unclear about usefulness of circular-ON switch described in this study due to sensitivity and cell-type-dependent IRES activity.

We appreciate the reviewer raising the point regarding the usefulness of the circRNA-ON switch given the intrinsic cell type-dependent activity of IRESs. We would like to further clarify the rationale and applicability of implementing circRNA-ON switches to execute customized cell-type classification function, logic-computation

function, and amplify IRES expression differences between cell types:

First, although the HCV IRES itself indeed showed a bit cell-type dependent activity variation, such variation is not enough for accurate cell-type classification task. Moreover, the unmodified IRES itself can only distinguish Huh7 from other cell lines that show low HCV IRES activity, and cannot be a customizable cell classifier to classify other cell line.

Second, The IRES itself cannot perform logic-computation function, which limit the applicability to execute complex function. In contrast, the programmability and modularity of circ-ON switches allow targeting diverse cell types by simple changes in miRNA binding sites and implementing basic logic-computation functions.

Third, even though the HCV IRES showed inherent translation activity variations across cell types, amplifying these differences via circRNA-ON switches can still be highly meaningful, for example to widen the therapeutic window of drug treatments. In our study, the basal LDH activity induced by GSDMD expression showed a 5.8-fold difference between Huh7 and HEK293 cells, but introducing circRNA-ON regulation expanded this difference to 18.6-fold (**Fig. 3g**). Also, although circRNA-OFF regulation expanded the difference to 51-fold, combining circRNA-OFF and circRNA-ON regulation amplified the difference to ~90-fold (**Fig. 3i**).

In summary, the usefulness of circRNA-ON switches is three-fold: (1) the programmable and customizable cell-type classification function; (2) the capacity to perform logic-computation function; (3) even though intrinsic IRES cell-type activity variations exist, further magnifying these differences using circRNA-ON switches still has practical usefulness and applicability.

Minor points

1. There are several grammatical errors in the manuscript. For example, “in presence/absence of” and “To further explored our MITAs.” The manuscript would benefit from language editing by a native English speaker.

We thank the reviewer for pointing out the grammatical errors in the manuscript. The current revised manuscript has been thoroughly reviewed and corrected for grammatical issues.

2. Line 83: Please correct “human embryo kidney” to “human embryonic kidney.”

We appreciate the reviewer catching the incorrect term "human embryo kidney" on line 83. We have corrected it to "human embryonic kidney" as suggested.

3. There are orthographic variants in miR-199a (e.g., miR199a, miRNA-199a-3p, miR-199a-3p). Please use consistent terminology.

We are grateful to the reviewer for noting the inconsistent terminology used for miR-199a. The terminology has now been standardized throughout the manuscript.

4. Please correct “in-vitro” to “in vitro.”

We are grateful to the reviewer for indicating the incorrect 'in-vitro' writing. The term 'in-vitro' has been corrected to '*in vitro*' in the manuscript.

5. Related to the previous minor point 4, the authors should carefully revise the manuscript. I think “Lipo8000” is the correct material rather than “Lipofectamine 8000”.

We appreciate the reviewer for pointing out the incorrect material name. The material name has been corrected to 'Lipo8000'.

Reviewer #2 (Remarks to the Author):

the authors addressed the reviewer's comments

We sincerely thank the reviewer for taking the time and effort to carefully review this manuscript.

REVIEWERS' COMMENTS

Reviewer #1 (Remarks to the Author):

The manuscript revised by Ning et al. has been improved through additional experiments and revisions. I think this work can be published after the following minor revisions.

1. The authors should mention and discuss the following points in the main text.

-IRES activity itself is cell-type dependent (the IRES itself has the potential to classify cell types).

-Detailed discussion of the usefulness of circular-ON switch in this study.

2. The figure number should be rearranged. The order should correspond to the order that appeared in the main text. (e.g. Supplementary Figure 6 currently appears after Supplementary Figure 9)

3. Except for Figure 4e, it is unclear what cell type was used in experiments. Please clarify them.

4. The legends for Supplementary figures could be positioned immediately after their respective figures.

5. Regarding Fig. 3j, k and Supplementary Fig. 8e-h, the authors described in Figure legend for Fig. 3 that they used noncytotoxic (control) circRNA as control, but the authors described that empty LNP was used as control in the main text. Which description is correct?

6. Line 165: Please correct "To ruled out" to "To rule out."

REVIEWERS' COMMENTS

Reviewer #1 (Remarks to the Author):

The manuscript revised by Ning et al. has been improved through additional experiments and revisions. I think this work can be published after the following minor revisions.

We sincerely appreciate the reviewer taking the time and effort to thoroughly review our revised manuscript and provide insightful feedback that helped improve the quality of our work. We have addressed the reviewer's concerns below and have incorporated these changes in our revised manuscript.

1. The authors should mention and discuss the following points in the main text.
 - IRES activity itself is cell-type dependent (the IRES itself has the potential to classify cell types).
 - Detailed discussion of the usefulness of circular-ON switch in this study.

We appreciate the reviewer's comments. We have added these points in the Discussion section.

2. The figure number should be rearranged. The order should correspond to the order that appeared in the main text. (e.g. Supplementary Figure 6 currently appears after Supplementary Figure 9)

We have rearranged the number of Supplementary figures accordingly. The previous Supplementary Figure 6,7,8,9 has been rearranged as new Supplementary Figure 9,6,7,8, corresponding to the order that appeared in the main text.

3. Except for Figure 4e, it is unclear what cell type was used in experiments. Please clarify them.

We have added information about the cell types used in the experiments in either panels or legends of Figures.

4. The legends for Supplementary figures could be positioned immediately after their respective figures.

We have revised the Supplementary Information and moved the legends for supplementary figure to the appropriate position.

5. Regarding Fig. 3j, k and Supplementary Fig. 8e-h, the authors described in Figure legend for Fig. 3 that they used noncytotoxic (control) circRNA as control, but the authors described that empty LNP was used as control in the main text. Which

description is correct?

We apologized for any confusion regarding to the control experiments. The control group in cell-type classifier circRNA transfection experiments was the noncytotoxic circRNA. We have corrected the text accordingly.

6. Line 165: Please correct “To ruled out” to “To rule out.”

We thank the reviewer for pointing out the grammatical error in the manuscript. We have corrected grammar errors throughout the manuscript.